# Analysis of 10,478 cancer genomes identifies candidate driver genes and opportunities for precision oncology

Ben Kinnersley [1,2,11], Amit Sud [1,3,4,5,6,11], Andrew Everall[1,11], Alex J. Cornish[1], Daniel Chubb[1], Richard Culliford[1], Andreas J. Gruber [7], Adrian Lärkeryd[8], Costas Mitsopoulos[9], David Wedge [10] & Richard Houlston [1] ✉

Tumor genomic profiling is increasingly seen as a prerequisite to guide the treatment of patients with cancer. To explore the value of whole-genome sequencing (WGS) in broadening the scope of cancers potentially amenable to a precision therapy, we analysed whole-genome sequencing data on 10,478 patients spanning 35 cancer types recruited to the UK 100,000 Genomes Project. We identified 330 candidate driver genes, including 74 that are new to any cancer. We estimate that approximately 55% of patients studied harbor at least one clinically relevant mutation, predicting either sensitivity or resistance to certain treatments or clinical trial eligibility. By performing computational chemogenomic analysis of cancer mutations we identify additional targets for compounds that represent attractive candidates for future clinical trials. This study represents one of the most comprehensive efforts thus far to identify cancer driver genes in the real world setting and assess their impact on informing precision oncology.

Precision oncology aims to tailor therapy to the unique biology of the patient's cancer, thereby optimizing treatment efficacy and minimizing toxicity[1,2]. Underpinning precision oncology is the concept of somatic driver mutations as the foundation of cancer biology[3,4].

The expansion in the number of therapeutically actionable genes has exposed the limitations of single-analyte genomic assays in cancer[5]. The modest incremental cost of adding additional cancer genes to high-throughput sequencing-based panels has made the development of drugs targeting increasingly smaller subsets of molecularly defined patients with cancer financially and logistically feasible[6]. The development of inhibitors effective in cancers driven by rare genomic mutations has required the concurrent development of clinical trial designs, such as basket trials, in which eligibility is

based on mutational status instead of organ site, cancer stage and histology[7]. With the advent of basket studies, many oncologists now consider that tumor genomic profiling should be offered to all patients with cancer who are not candidates for curative-intent local or systemic therapy[8].

At present, several standalone tests or a panel are typically used to capture a set of genomic, transcriptomic or epigenomic features in a tumor to inform patient treatment[9]. However, falling costs are making whole-genome sequencing (WGS) a potentially attractive proposition as a single all-encompassing test to identify cancer drivers and other genomic features, which may not be captured by standard testing but are clinically actionable[10]. This approach is being explored in the UK by the 100,000 Genomes Project (100kGP), which is seeking to deliver the

[1]Division of Genetics and Epidemiology, The Institute of Cancer Research, London, UK. [2]University College London Cancer Institute, University College London, London, UK. [3]Department of Medical Oncology, Dana-Farber Cancer Institute, Boston, MA, USA. [4]Broad Institute of MIT and Harvard, Cambridge, MA, USA. [5]Harvard Medical School, Boston, MA, USA. [6]Centre for Immuno-Oncology, Nuffield Department of Medicine, University of Oxford, Oxford, UK. [7]Systems Biology & Biomedical Data Science Laboratory, University of Konstanz, Konstanz, Germany. [8]Division of Molecular Pathology, The Institute of Cancer Research, London, UK. [9]Division of Cancer Therapeutics, The Institute of Cancer Research, London, UK. [10]Manchester Cancer Research Centre, University of Manchester, Manchester, UK. [11]These authors contributed equally: Ben Kinnersley, Amit Sud, Andrew Everall. ✉e-mail: richard.houlston@icr.ac.uk

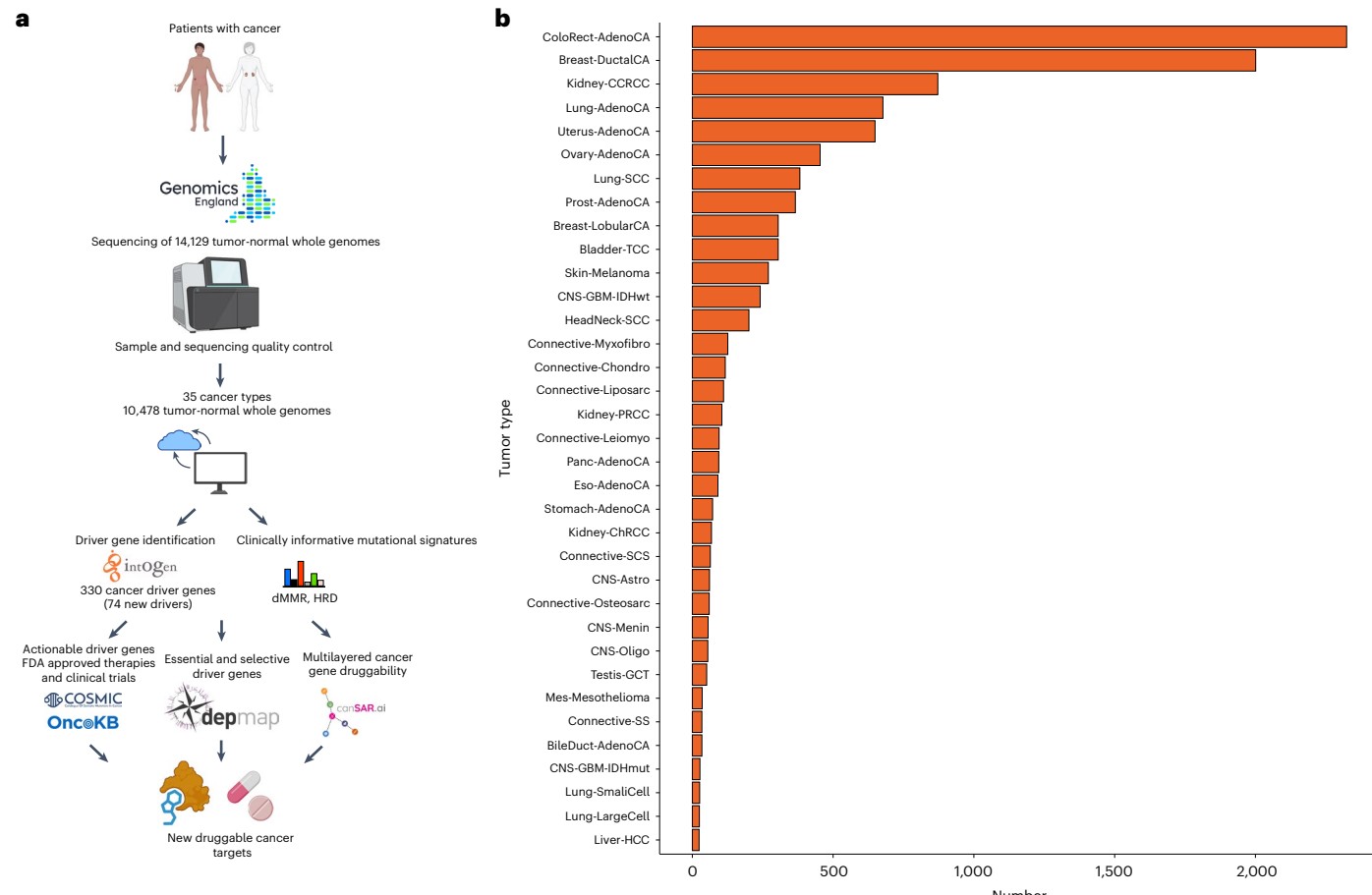

**Fig. 1 | Study design and number of samples per tumor type included in the analysis. a**, Study design. **b**, Number of samples per tumor type. BileDuct-AdenoCA, bile duct adenocarcinoma; Bladder-TCC, bladder transitional cell carcinoma; Breast-DuctalCA, breast ductal carcinoma; Breast-LobularCA, breast lobular carcinoma; CNS-Astro, astrocytoma; CNS-GBM-IDHmut, *IDH* mutated glioblastoma; CNS-GBM-IDHwt, *IDH* wild-type glioblastoma; CNS-Menin, meningioma; CNS-Oligo, oligodendroglioma; ColoRect-AdenoCA, colorectal adenocarcinoma; Connective-Chondro, chondrosarcoma; Connective-Leiomyo, leiomyosarcoma; Connective-Liposarc, liposarcoma; Connective-Myxofibro, myxofibrosarcoma; Connective-Osteosarc, osteosarcoma; Connective-SCS, spindle cell sarcoma; Connective-SS, synovial sarcoma; Eso-AdenoCA,

esophageal adenocarcinoma; HeadNeck-SCC, squamous cell carcinoma of the head and neck; Kidney-CCRCC, clear cell renal cell carcinoma; Kidney-ChRCC, chromophobe renal cell carcinoma; Kidney-PRCC, papillary renal cell carcinoma; Liver-HCC, hepatocellular carcinoma; Lung-AdenoCA, lung adenocarcinoma; Lung-LargeCell, large cell lung cancer; Lung-SCC, squamous cell carcinoma of the lung; Lung-SmallCell, small cell carcinoma of the lung; Mes-Mesothelioma, mesothelioma; Ovary-AdenoCA, ovarian adenocarcinoma; Panc-AdenoCA, pancreatic adenocarcinoma; Prost-AdenoCA, prostate adenocarcinom; Skin-Melanoma, melanoma of the skin; Stomach-AdenoCA, gastric adenocarcinoma; Testis-GCT, testicular germ cell tumor; Uterus-AdenoCA, uterine adenocarcinoma. Fig. 1a created with BioRender.com.

vision of precision oncology through WGS to National Health Service (NHS) patients as part of their routine care[11].

Here, we report an analysis of WGS data on 10,478 patients spanning 35 cancer types recruited to the 100kGP (Fig. 1a). Across all cancer types we identify 330 candidate driver genes, including 74 which are new to any cancer. We relate these to their actionability both in terms of currently approved therapeutic agents and through computational chemogenomic analysis to predict candidacy for future clinical trials.

## Results

We analysed 10,478 cancer genomes spanning 35 different cancer types (Fig. 1b and Supplementary Tables 1 and 2). While broadly reflecting the spectrum and frequencies of cancers diagnosed in the UK population, there were differences, with an over-representation of colorectal and kidney cancers and a paucity of prostate and pancreatic cancers (Extended Data Fig. 1). Additionally, for the main cancer types, the patients recruited to 100kGP tended to be younger and had earlier stage tumors compared to patients in the general UK population (Supplementary Table 3).

Mutation rates varied across the different cancer types with cutaneous melanoma having the highest single nucleotide variant mutation count and meningioma the lowest (Extended Data Fig. 2). A total of 945 samples, notably colorectal and uterine cancers, were hypermutated, either as result of defective mismatch repair (dMMR) or *POLE* mutation. Invasive ductal carcinoma of the breast had the highest power for driver gene detection (>90% power for a mutation rate of at least 2% higher than background) and large cell lung cancer had the lowest power (Fig. 2 and Supplementary Table 4). Compared with the recent Pan-Cancer Analysis of Whole Genomes analysis[12], the 100kGP cohort was better powered to identify a driver mutation for 19 cancers, notably for breast, colorectal, esophageal and uterine cancer, lung adenocarcinoma and bladder transitional cell carcinoma where the sample sizes were more than tenfold higher.

### Spectrum of cancer driver genes

Across all cancer types we identified 770 unique tumor–driver gene pairs corresponding to 330 unique candidate cancer driver genes (Fig. 3, Extended Data Fig. 3 and Supplementary Table 5). When

compared to the largest pan-cancer driver analysis, in 21 of 31 cancer types where tumor histologies could be matched, we recovered 61% of all cancer drivers reported by the Catalogue of Somatic Mutations in Cancer (COSMIC), the Integrative OncoGenomics (IntOGen)[4] and The Cancer Genome Atlas (TCGA) Program pan-cancer analysis reported by ref. 13 (Supplementary Table 5). We were able to detect 80% of drivers reported for colorectal, breast, lung and ovarian cancers but only <20% of drivers reported for hepatocellular and stomach cancers, which may be a result of differing sample size or intertumour heterogeneity[14]. The number of identified cancer driver genes varied between cancer types, with colorectal and uterine cancers having the most (60 genes) and spindle cell carcinoma having the fewest (4 genes). Across the 35 cancers, we found no correlation between average mutation burden and the number of driver genes in each cancer (Pearson's $r$ = 0.19, $P$ = 0.27). The consensus list also includes 326 tumor–driver pairs that have not previously been reported by the Cancer Gene Census, IntOGen or the pan-cancer analysis of TCGA[4,13] (Supplementary Table 5) and 74 that have not previously been associated with any specific tissue. Almost all of the candidate drivers identified were uncommon, with 88% (65 of 74) having a mutation frequency <10% in the respective cancer type. The highest numbers of new cancer driver genes were found for uterine ($n$ = 42), bladder ($n$ = 40) and colorectal ($n$ = 37) cancers. Furthermore, we identified drivers in tumor types which have not been cataloged by IntOGen[4] and ref. 13. These include breast lobular carcinoma, meningioma and myxofibrosarcoma. Predictions of known cancer driver genes in new cancer types include *SPTA1*, *CHD4* and *ASXL1* in colorectal cancer, *FOXO3*, *MUC16* and *ZFPM1* in breast cancers and *CNTNAP2*, *CTNND2* and *TRRAP* in lung adenocarcinoma. Entirely new predictions include *MAP3K21* (encoding a mixed-lineage kinase) in colorectal cancer, *USP17L22* (encoding a deubiquitinating enzyme) in breast ductal carcinoma and *TPTE* (encoding a tyrosine phosphatase) in lung adenocarcinoma (Supplementary Table 5).

Eighty-five genes were identified as a driver in more than two tumor types, with 26 genes functioning as drivers in more than five tumor types (Fig. 4a). As expected, *TP53* was identified as a driver gene in the most tumor types, followed by *PIK3CA*, *ARID1A* and *PTEN*, acting as cancer driver genes in 29, 18, 16 and 14 different tumor types, respectively. While many genes function as drivers in several cancer types, some drivers are mutated at high frequencies only in specific tumors, such as *VHL* in clear cell renal cell carcinoma and *FGFR3* in bladder cancer (Fig. 4a). Across drivers operating in several cancer types, the clearest examples of domain-specific driver mutations were in *EGFR*, where protein tyrosine and serine/threonine kinase domain mutations predominated in lung adenocarcinoma, in contrast to extracellular furin-like cysteine-rich region domain mutations in *IDH* wild-type glioblastoma (Supplementary Table 6 and Extended Data Fig. 4a). *PIK3CA* also showed a preference for p85-binding domain mutations in uterine adenocarcinoma compared to other cancer types, such as breast ductal carcinoma, which are enriched for mutations in the PIK family domain (Supplementary Table 6 and Extended Data Fig. 4b). Hierarchical clustering of cancers based on the presence of identified driver mutations and their respective $q$ value demonstrated clustering of cancer types by cell of origin (for example, head and neck and lung squamous cell carcinoma) and by organ (for example, breast ductal and lobular carcinomas; Extended Data Fig. 5). The ratio of predicted activating versus tumor suppressor driver genes varied across tumor types with meningioma and myxofibrosarcoma possessing the highest and lowest ratios, respectively (Fig. 4b and Supplementary Table 5).

Across the 35 different tumor types in 9,070 unique samples we identified 12,606 distinct oncogenic mutations in tumor-relevant cancer driver genes. The median number of oncogenic mutations in cancer driver genes per sample was two, across all tumors. The highest median number of oncogenic mutations in driver genes per sample was seen in uterine cancer ($n$ = 6; Extended Data Fig. 6). We observed significant differences ($P_{binomial}$ < 3.5 × 10$^{-3}$) in oncogenic mutation

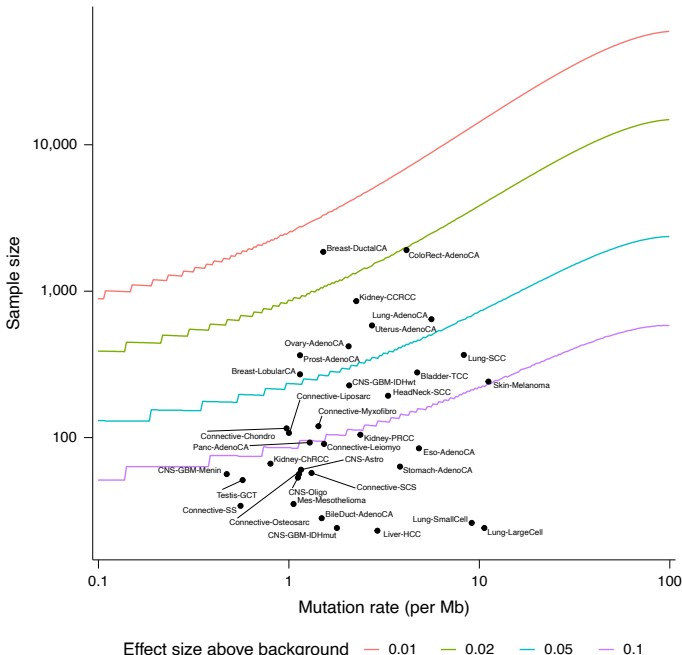

**Fig. 2 | Power estimates for driver gene identification per tumor type.** The number of samples needed to achieve 90% power for 90% of genes ($y$ axis). Gray vertical lines indicate exome-wide background mutation rates ($x$ axis). Black dots indicate sample sizes and mutation rates in the current study.

frequency in cancer driver genes across different tumor histologies arising from the same organ. Examples include *CDH1*, *TBX3* and *TP53* in breast cancers, *ATRX*, *CIC*, *IDH1*, *PTEN* and *TP53* in central nervous system tumors, *IDH1* and *TP53* in connective tissue tumors, *PBRM1* and *VHL* in renal cancers and *EGFR*, *KMT2D*, *KRAS*, *NFE2L2*, *PTEN*, *STK11* and *TP53* in lung cancers (Fig. 5).

Considering all 330 cancer driver genes, 217 featured at least one clonal oncogenic mutation (214 clonal, 167 clonal early and 114 clonal late events (Supplementary Table 7). *APC*, *TP53* and *PIK3CA* possessed the most clonal oncogenic mutations (Fig. 6a and Extended Data Fig. 7). Of the 162 driver genes that harbored at least one subclonal oncogenic mutation, *ARID1A*, *TP53* and *PIK3CA* possessed the most (Fig. 6b and Extended Data Fig. 7). Consistent with published work, a high proportion (55%) of all early clonal driver mutations occur in just four genes (*TP53*, *APC*, *KRAS* and *PIK3CA*) whereas the equivalent percentage of late and subclonal oncogenic mutations was observed in 19 different genes (Supplementary Table 7)[15-18]. This finding supports a model in which early events in cancer evolution tend to occur in a restricted set of driver genes and a wider range of drivers feature late in tumor evolution. In tumors with more than ten oncogenic mutations, meningioma exhibited the greatest proportion of clonal oncogenic mutations (Extended Data Fig. 8a). Large cell lung, testicular germ cell tumor and oligodendroglioma carried the highest proportion of early clonal, late clonal and subclonal oncogenic mutations, respectively (Extended Data Fig. 8b–d).

### Sensitivity of WGS mutation detection compared to panels

We initially investigated the performance of WGS to detect clinically relevant mutations compared to conventional panel-based testing through comparison of mutation calls with Memorial Sloan Kettering (MSK) Cancer Center cohorts at 43 established drivers (Supplementary Note 1). For primary tumors represented in the MSK and 100kGP cohorts, the rate of mutations called for each driver gene was comparable (Supplementary Figs. 1 and 2). Thereafter, we estimated the sensitivity of mutation detection in the 100kGP cohort by extracting

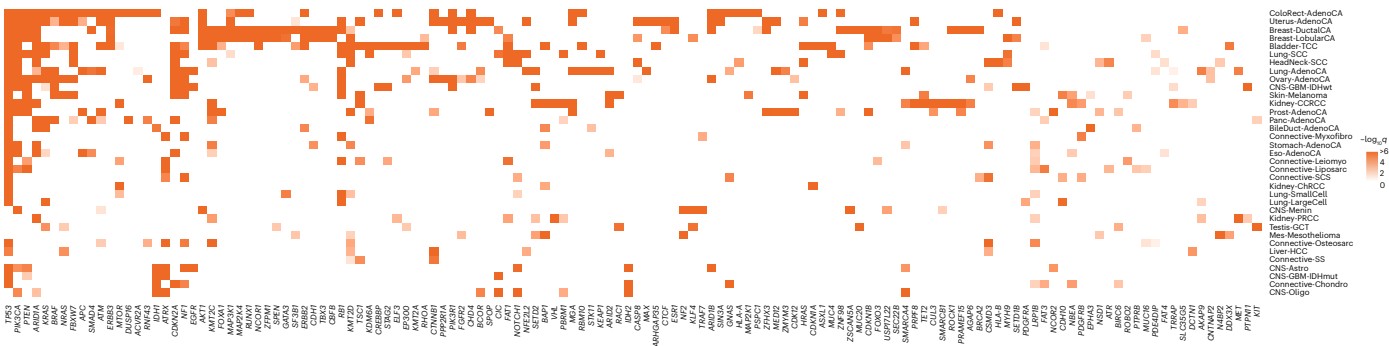

**Fig. 3 | Heatmap of candidate cancer driver genes identified in at least two different cancer types.** Heatmap intensity proportional to $q$ value.

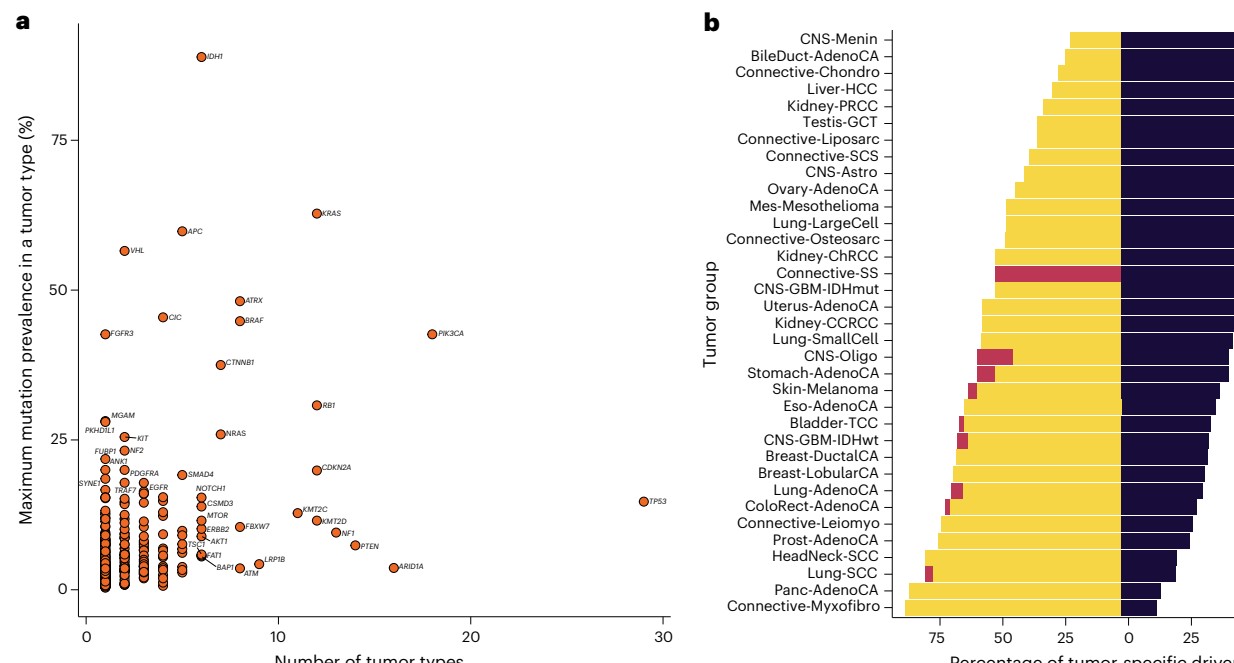

**Fig. 4 | Distribution and predicted function of candidate cancer driver genes across tumor types. a**, Distribution of driver genes across different types of cancer: $y$ axis, maximal mutational prevalence in a tumor type; $x$ axis, number of tumor types in which the driver gene is identified. Genes labeled are candidate drivers in at least six tumor types or have a maximum mutation prevalence in a tumor type of >17%. **b**, Distribution of cancer driver gene function associated with each cancer type: $y$ axis, tumor group; $x$ axis, percentage of tumor-specific driver genes.

per-tumor coverage across the panel of 43 driver genes (Supplementary Note 1). Specifically, for 88% of cancer driver genes, the expected sensitivity for mutation detection was >99% in the 100kGP cohort. Furthermore, for 90% of cancer driver genes, >98% of the coding sequence had sufficient coverage such that more than six reads could be used for mutation detection after accounting for tumor purity (Supplementary Figs. 3–7). These findings are in agreement with published data on the diagnostic accuracy of 100kGP WGS compared to panel sequencing conducted by Genomics England (sensitivity of 99% for variant allele frequency >5% and coverage >70×).

### Actionability of driver gene mutations

We next sought to evaluate the landscape of clinically actionable driver alterations through reference to the COSMIC and Precision Oncology Knowledge Base (OncoKB). We observed that both the fraction of samples and proportion of alteration types varied across tissue types. Data from COSMIC indicated that 85% of all samples (8,880 of 10,478) possessed at least one putatively actionable alteration being targeted in a clinical setting (Fig. 7a and Supplementary Table 8), while 55% of samples (5,805 of 10,478) had at least one putatively actionable or

biologically relevant alteration from OncoKB (Fig. 7b and Supplementary Tables 9 and 10). Across all cancer types, 15% (1,560 of 10,470) of the patients would be eligible for a currently approved therapy as defined by OncoKB. Of the actionable mutations defined by OncoKB ($n$ = 9,639), 5,823 were clonal, 2,632 were early clonal, 229 were late clonal and 852 were subclonal.

The most common putatively actionable alterations across all of the 35 cancer types were mutations in *PIK3CA*, *KRAS* and *PTEN* (Supplementary Fig. 8). *PIK3CA* encodes the p110α protein, which is a catalytic subunit of phosphatidylinositol 3-kinase (PI3K). Specific oncogenic missense mutations in *PIK3CA* were present in 50% of lobular breast cancers and 38% of ductal breast cancers and their presence is an indication for the use of PI3Kα inhibitor alpelisib[19]. These mutations are present in a number of cancers including colorectal (20%) and uterine cancers (47%) and in these tumor types are subject to early clinical studies with an allosteric inhibitor of PI3Kα[20]. We found high fractions of patients with pancreatic cancer, colorectal cancer and lung adenocarcinoma with actionable *KRAS* mutations (39–64% of all cases). The *KRAS* G12C mutation was present in 17% of lung adenocarcinoma cases and is targeted by mutation-specific selective covalent inhibition with adagrasib

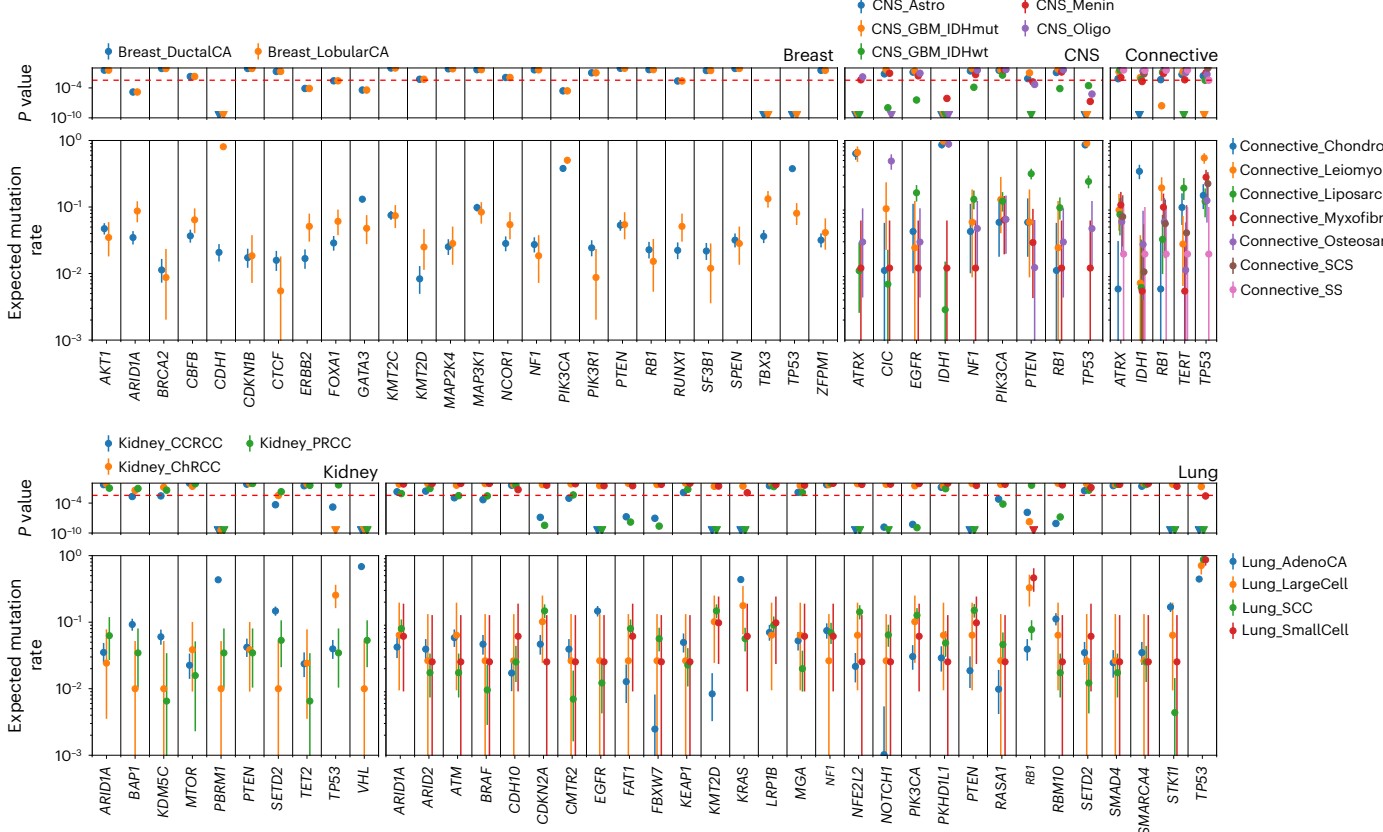

**Fig. 5 | Comparison of driver gene somatic mutation rates between tumor histologies.** Expected mutation rate and 95% confidence intervals of each driver in the cohort (2,306 breast, 440 central nervous system (CNS), 1,045 kidney, 1,110 lung and 607 connective tissue tumors in the 100kGP cohort) based on the number of samples in which the driver gene is mutated for the given tumor histology. Binomial *P* values are shown. The dashed red line corresponds to a false discovery rate of 0.01.

or sotorasib[21,22]. PI3Kβ inhibition is of significant biological interest in patients with oncogenic inactivating *PTEN* mutations, as PI3Kβ is thought to drive cellular proliferation in these tumors. Inactivating *PTEN* mutations were prevalent in melanoma (10%), hepatocellular carcinoma (13%), squamous cell carcinoma of the lung (15%), glioblastoma multiforme (29%) and uterine carcinoma (66%) and their presence would result in eligibility for early studies of PI3Kβ inhibition[23].

### Landscape of clinical actionability
In addition to actionable mutations in single genes, other classes of molecular alterations are recognized as tumor-agnostic biomarkers of drug response. These include mutational profiles caused by dMMR/*POLE* mutations and homologous recombination deficiency (HRD), which represent phenotypic markers for response to immunotherapy and PARP inhibition respectively. A total of 319 tumors (3%) exhibited a mutational signature for HRD, which provides an indication for PARP inhibition therapy and potential sensitivity to platinum chemotherapy[24–28]. As demonstrated in our companion paper, the etiological basis of HRD was, however, only identifiable in 16% of these cases based on biallelic inactivation of *BRCA1*, *BRCA2*, *PALB2*, *BRIP1* or *RAD51B* through germline and somatic mutations[29]. While other cases may be caused by promoter methylation, which could not be assessed because these data are not available for 100kGP samples, the findings provide a strong rationale for extending the number of patients potentially eligible for PARP inhibitors rather than solely relying on BRCA-testing. A total of 1,309 tumors possessed a high coding tumor mutational burden (more than ten mutations per megabase, Mb) and 144 cancers had evidence of dMMR. Considering these collectively would suggest that 1,312 patients may be eligible for checkpoint inhibition[30,31]. To

explore the prospect of several targeted therapies being used in the same patient, we combined the OncoKB clinical actionability annotations with that of TMB, dMMR and HRD clinical actionability annotations. In total, 11,503 independent unique gene targets were present in 6,151 samples with 34% (3,577 of 10,478) of tumors possessing one, 13% (1,361 of 10,478) two and 12% (1,213 of 10,478) possessing at least three clinically actionable driver mutations.

### Expanding the druggable cancer genome
An opportunity emerging from the systematic analysis of cancer genomes is the identification of new therapeutic intervention strategies. Of the 330 candidate cancer driver genes identified in this study, 261 (79%) are not currently identified as therapeutic targets in either COSMIC or OncoKB databases. As a means of triaging these genes as candidates for therapeutic intervention, we assessed the essentiality and selectivity of driver genes and their druggability using RNAi/CRISPR DepMap data and the integrative cancer-focused knowledge-base, canSAR, respectively[32,33]. We found 96 of 261 (37%) of these genes are predicted to be commonly essential and of these 12 of 96 (13%) have a chemical probe available and 35 of 96 (36%) have a ligandable three-dimensional (3D) structure (Supplementary Table 11).

Motivated by the observation that targeting proteins which interact with cancer driver genes can result in successful precision oncology strategies, we sought to expand the network of druggable targets in cancer[34,35]. To this end, we used canSAR to map and pharmacologically annotate networks of the cancer genes identified for each tumor type. Specifically, we seeded networks with driver genes identified in each tumor group and used transcriptional and curated protein–protein interactions to recover a refined cancer-specific network of proteins,

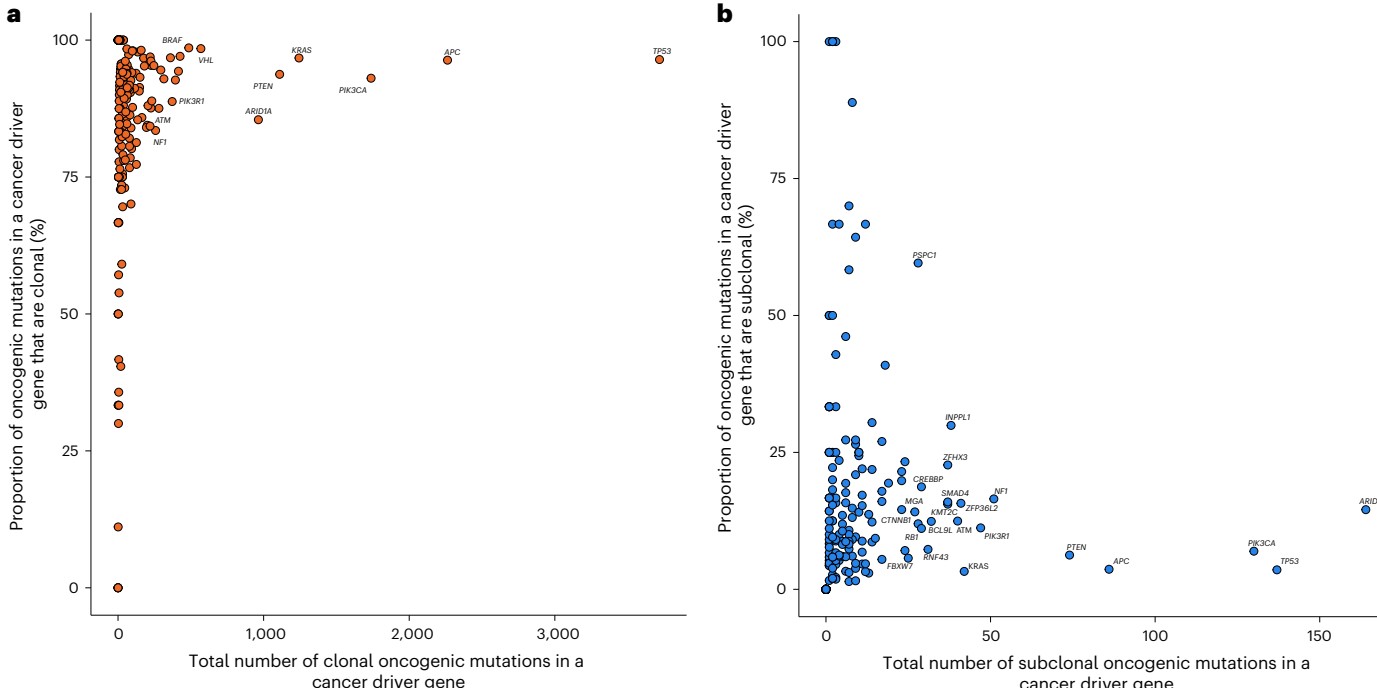

**Fig. 6 | Distribution of clonal and subclonal oncogenic mutations in candidate cancer driver genes. a**, Distribution of clonal oncogenic mutations in candidate cancer driver genes across all cancer types: *y* axis, percentage of all clonal oncogenic mutations of all oncogenic mutations; *x* axis, total number of clonal oncogenic mutations. Clonal oncogenic mutations include clonal mutations that occurred before duplications involving the relevant chromosome (early), clonal mutations that occurred after such duplications (late), and mutations that occurred when no duplication was observed. Genes labeled are those with >250 clonal oncogenic mutations or clonal oncogenic mutations represent >95% of all oncogenic mutations. **b**, Distribution of all subclonal oncogenic mutations in candidate cancer driver genes across all cancer types: *y* axis, percentage of all subclonal oncogenic mutations of all oncogenic mutations; *x* axis, total number of subclonal oncogenic mutations. Genes labeled are those with >50 subclonal oncogenic mutations and >5% of all oncogenic mutations as subclonal.

each protein being annotated on the basis of several assessments of 'druggability', that is the likelihood of the protein being amenable to small molecule drug intervention. After seeding each cancer-specific network with their respective drivers, we yielded a total of 631 distinct proteins across all cancers (Supplementary Table 12). The median number of unique proteins in each network across all cohorts was 57, with colorectal cancer possessing the largest network (*n* = 231; Extended Data Fig. 9) and spindle cell carcinoma possessing the smallest network (*n* = 10). As expected there was a correlation between network size and number of identified drivers for each cancer type (Pearson's *r* = 0.9, *P* = 1.23 × 10⁻⁹).

Of these 631 proteins, 58% (*n* = 369) were retrieved solely through network analysis, of which most (*n* = 323) were not formally identified as candidate driver genes in any cancer type (hereafter referred to as cancer-network proteins). Notable examples include *HDAC1*, *CDK2* and *CDK1*, which were present in 31, 29 and 28 cohorts, respectively. We observed 70% (*n* = 225) of these cancer-network proteins as being targetable by existing approved or investigational therapies, with notable examples including *BCL2* and *BTK*. Of the remaining 97 genes, 34 are commonly essential, 11 possess concordant lineage specificity, 48 are ligandable by 3D structure and 11 have an existing high-quality probe available (Supplementary Table 13). Collectively these data provide potential future opportunities for therapy for several cancers. For example, *CDC5L*, a core component of the Prp19 (hPrp19)/Cdc5L pre-RNA splicing complex, is part of the melanoma cancer protein network[36]. This protein is predicted to be commonly essential with lineage specificity and has a 3D ligandable structure.

## Discussion

Clinical and laboratory observations have led to the recognition that genomic profiling of tumors is increasingly important for the management of patients with cancers[37]. To explore the value of WGS to precision oncology we have analysed WGS data on 10,470 patients recruited to the 100kGP study.

Across all cancers, we identified 330 cancer driver genes, 74 of which are new to any cancer type. The candidate driver gene list is limited by focusing on point mutations and small indels without consideration of copy-number alterations, genomic fusions or methylation events. Nevertheless, we believe it represents one of the most comprehensive efforts thus far to identify cancer driver genes and serves as an important research asset. The similarities and differences in driver mutation frequencies in cancers arising from the same organ imply both shared and divergent pathways in oncogenesis. Notably, however, many driver mutations are common across several different tumor types. If clinically translated, these observations suggest that currently 55% of patients' tumors harbor a potentially actionable mutation, either in terms of predicting sensitivity to certain treatments or clinical trial eligibility. This contrasts with 22% achievable if based on the current small variant testing panels in widespread use[38]. Although our assumption is predicated on approved drugs as a proxy for effective cancer therapies, a recent study of cancer drug approvals by the Food and Drug Administration (FDA) concluded that new cancer drug approvals reduce the risk of death and tumor progression[39]. To inform potential future therapeutic opportunities, we applied established chemogenomic technologies to map and pharmacologically annotate the cellular network of cancer genes identified by WGS. Through annotation of cellular networks with measures of essentiality and selectivity, we were able to highlight additional potential therapeutic targets in cancer. It is likely that such endeavors will be improved by exploiting emergent high-throughput reporter assays to assess the functional consequences of somatic driver mutations in greater detail[40].

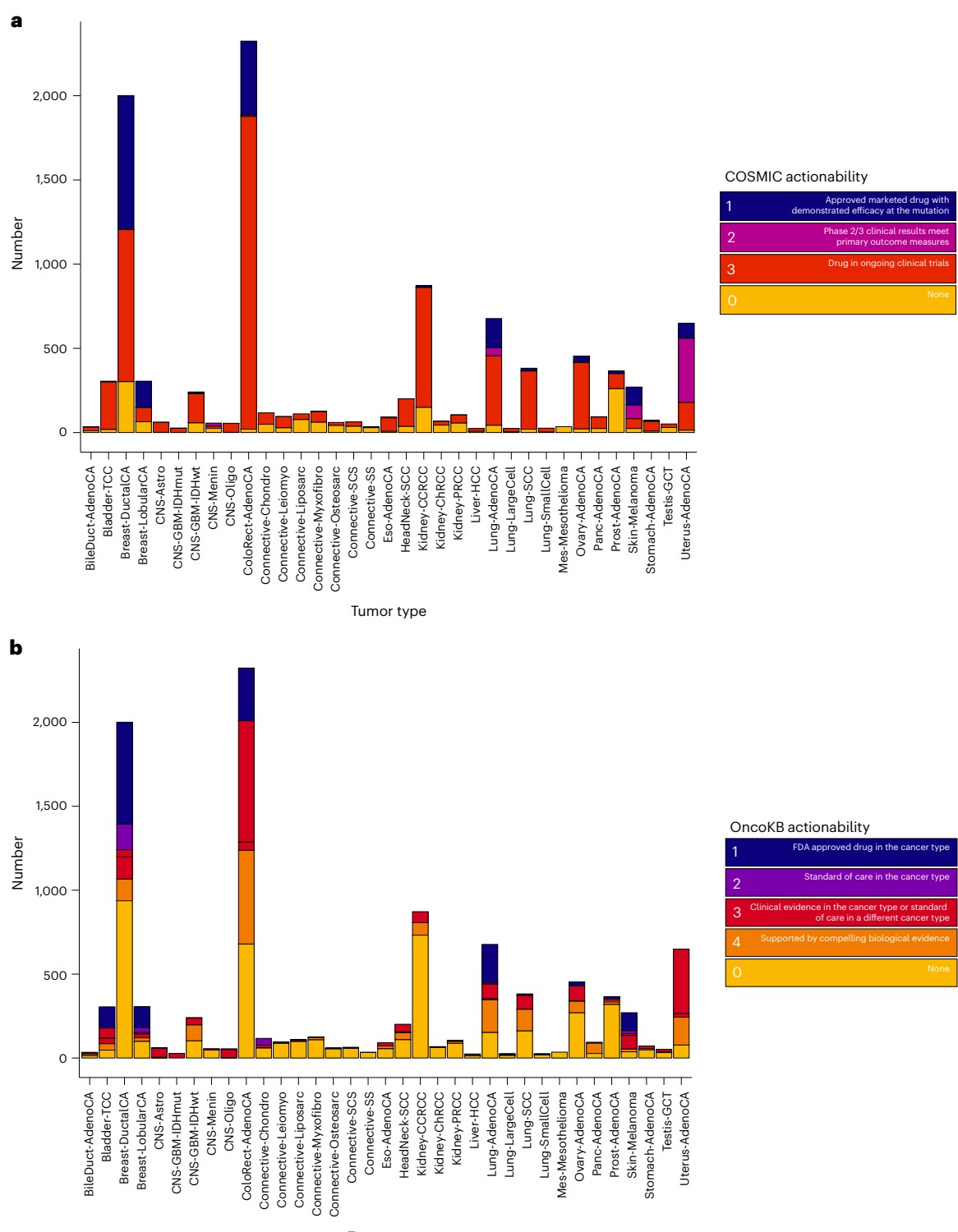

**Fig. 7 | Clinical actionability ascribable to each candidate cancer driver gene. a**, Clinical actionability ascribable to each candidate cancer driver gene according to COSMIC by cancer type. Tumors were annotated by the highest scoring gene mutation–indication pairing, with 'None' indicating no actionable mutations were detected in the tumor. **b**, Clinical actionability ascribable to each candidate cancer driver gene according to OncoKB by cancer type. Tumors were annotated by the highest scoring gene mutation–indication pairing, with 'None' indicating no actionable mutations were detected in the tumor.

The strengths of this study not only include the cohort size but the combination of systematic processing of samples and data arising from several treatment centers across England. These strengths minimize the impact of between-center sequencing effects while ensuring a representative cohort of cancers are captured[41]. We do,

however, acknowledge that while the spectrum of cancers included in our analysis is largely representative of those diagnosed in the United Kingdom, patients recruited to 100kGP are younger and predominantly have early-stage disease. Furthermore, characteristics such as patient ancestry and geography can affect the mutagenic profile of tumors,

which potentially impacts on the generalizability of our findings to worldwide populations[42,43].

Accepting these limitations, our observations indicate that, depending on cancer type, approximately 15% of patients are potentially eligible for a currently approved therapy targeting an oncogenic driver. Our discovery analysis, however, implies that far more patients may potentially be candidates for a therapy targeting a driver mutation or pathway. A long-standing criticism of precision oncology is that often its proponents overstate the clinical actionability of individual genes or genomic variants[44]. Mutations that are clinically validated and FDA-recognized as predictive biomarkers of drug response are often grouped together as clinically actionable, with such mutations potentially erroneously identified as the putative basis for outlier exceptional responses. To better communicate the strength of evidence supporting the clinical actionability of individual mutant alleles, many variant knowledge bases stratify genomic alterations on the basis of the level of clinical and/or biological data supporting their use as a predictive biomarker of drug response or resistance. Here, we have sought to address such concerns by making use of well-curated resources to assign actionability to driver mutations. Specifically, we have queried knowledge databases which are regularly curated by an expert panel and are therefore recognized to reflect the current state of knowledge[31].

While the 100kGP was predicated on delivering diagnostic tests for well-established actionable mutations in NHS cancer patients with high sensitivity, concern has been raised over missing well-recognized clinically actionable mutations[45]. In our analysis the frequency of established cancer-specific oncogenic drivers recovered was, however, comparable to MSK-IMPACT and MSK-MET[6,9]. Moreover, the sensitivity of 100× WGS to identify mutations was high even for samples with low tumor purity (Supplementary Note 1 and Supplementary Figs. 3–7).

A barrier to the broader success of precision oncology paradigms may be the many 'undruggable' oncogenic mutations coupled with the fact that targeting downstream effectors typically fails to demonstrate the levels of clinical efficacy of drugs that directly inhibit the mutated oncoprotein. Recent developments in protein structure prediction, new degraders, covalent inhibition and allosteric protein domain maps seek to unlock these 'undruggable' proteins[46–49]. Furthermore, WGS allows for the extension of analyses beyond the consideration of individual genetic alterations, thereby affording a clinically significant benefit over targeted panel sequencing assays. Mutational signatures associated with dMMR and HRD are increasingly being shown to be clinically relevant to defining responsiveness to immunotherapy and PARP inhibition, respectively[24,30]. Additionally, there is increasing evidence that other signatures reflecting the DNA repair capacity of cancer cells are predictive of drug responsiveness to other agents[5,50]. A more detailed discussion and comprehensive description of all classes of mutational signatures observed across the 100kGP are reported in our companion paper[29]. The ability to robustly characterize mutational signatures may therefore prove to be a major clinically significant incremental benefit of WGS over targeted panel sequencing assays. Moreover, the provision of WGS is likely to play a greater role in patient management given that T cell-based therapies are of increasing importance and in silico approaches are now used to predict the presence of immunogenic tumor-specific neoantigens from WGS[51–54].

Despite the merits of WGS as a one-stop clinical assay, its wider adoption outside selected academic and commercial centers has been limited[37]. A great hurdle is that the tumor material available for many patients is of insufficient quantity, quality or purity for these broader sequencing platforms. Indeed, in the 100kGP the lack of access to fresh frozen samples (and/or those of sufficient quantity) precluded the analysis of tumors from many patients[11]. In designing clinical assays, the limitations imposed by cost and sequencing capacity require the balancing of sequencing breadth and depth[41]. At present, the higher coverage of targeted assays represents an advantage over WGS for detection of alterations in genes clinically validated as biomarkers of drug response, especially in samples with poor DNA quality or high stromal contamination. A wider adoption of WGS will require further reductions in sequencing costs and technological improvements to enable the use of lower-quality, archival formalin-fixed, paraffin-embedded tumor tissue[55]. Any such developments will have to address the issue that formalin fixation adversely affects DNA quality and the ability to reliably call variants from WGS data, even when using bioinformatic correction[41,56,57]. Aside from such technical issues there are also inherent limitations to short-read WGS. Notably, structural variants cannot be robustly called, with low concordance being a feature of present implemented algorithms[58,59]. It is likely that this limitation will only be addressed by adoption of long-read sequencing, albeit presently this incurs a high requirement for DNA and further cost, thus restricting its use in the diagnostic setting[60]. The continued decline in sequencing costs and the identification of new genomic biomarkers predictive of drug response have driven the rapid adoption of multigene profiling of patients as a component of routine cancer care. As our analysis indicates, the future adoption of WGS or broader panels has the potential to enable more accurate assessments of the driver mutational landscape predictive of drug response.

## Online content

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

## Methods

### The 100kGP cohort

The analysed cohort comprised tumor–normal sample pairs from patients with primary cancers recruited to 100kGP (v.11 release) through 13 Genomic Medicine Centers across England (Supplementary Fig. 9). Genomics England has obtained written informed consent from all participants. We restricted our analysis to high-quality sequencing data derived from PCR-free, flash-frozen primary solid tumor samples from 10,470 adults (34 bile duct, 305 bladder, 2,306 breast, 2,324 colorectal, 440 central nervous system, 91 esophageal, 201 head and neck, 1,045 renal cell, 24 liver, 1,110 lung, 35 mesothelioma, 607 soft tissue, 454 ovarian, 94 pancreas, 366 prostate, 270 melanoma, 72 gastric, 51 testicular and 649 uterus) (Supplementary Tables 1–3). Comprehensive clinicopathology information on the patients is provided in Supplementary Table 3 and complete details on sample curation, tumor purity per cancer type (Extended Data Fig. 10), WGS, somatic variant calling, mutation annotation and power calculations are provided in Supplementary Note 1. We identified mutational signatures associated with dMMR and HRD in tumors using SigProfilerExtractor complemented by mSINGS and HRDetect (Supplementary Note 1)[29,61,62].

### Identification and timing of driver genes

Cancer driver genes for each of the tumor types were identified using the IntOGen pipeline (Supplementary Note 1)[4]. We examined the sensitivity of WGS in the 100kGP cohort to detect mutations in well-established driver genes based on sample purity and gene coverage and by comparing the call rates of panel sequencing reported in the Integrated Mutation Profiling of Actionable Cancer Targets and Metastatic Events and Tropisms studies of cancer conducted by the MSK Cancer Center (Supplementary Note 1)[6,63]. The relative evolutionary timings of candidate driver mutations were obtained using MutationTimeR (Supplementary Note 1)[15].

### Actionability of driver gene mutations and networks

We first queried the OncoKB and COSMIC Mutation Actionability in Precision Oncology Product databases to evaluate the therapeutic implications of genetic events[31,64]. Both databases catalog approved marketed drugs having demonstrated efficacy in tumors with specified driver gene mutations, based on clinical trials and published clinical evidence. OncoKB also provides compelling biological evidence supporting the cancer driver gene as being predictive of a response to a given drug.

To undertake a chemogenic analysis of cancer networks for each cancer type, we used protein products of the cancer driver genes to seed a search for all interacting proteins in the canSAR interactome[33], which is based on information from eight databases, including the IMeX consortium[65], Phosphosite[66] and key publications. We annotated proteins with pharmacological and druggability data using canSAR's Cancer Protein Annotation Tool. Essential and selective genes including lineage specificity were ascertained from the ShinyDepMap analysis server (Supplementary Note 1)[32].

### Reporting summary

Further information on research design is available in the Nature Portfolio Reporting Summary linked to this article.

## Data availability

Summary statistics for each tumor group are provided in the Supplementary Tables where such data do not enable identification of participants. All sample-specific WGS data and processed files from the 100,000 Genomes Project can be accessed by joining the Pan Cancer Genomics England Clinical Interpretation Partnership (GeCIP) Domain once an individual's data access has been approved (https://www.genomicsengland.co.uk/research/pan-cancer-and-molecular-oncology-community). The link to becoming a member of the Genomics England research

network and obtaining access can be found at https://www.genomicsengland.co.uk/research/academic/join-gecip. The process involves an online application, verification by the applicant's institution, completion of a short information governance training course and verification of approval by Genomics England. Please see https://www.genomicsengland.co.uk/research/academic for more information. The Genomics England data access agreement can be obtained from figshare at https://doi.org/10.6084/m9.figshare.4530893.v7 (ref. 67). All analysis of Genomics England data must take place within the Genomics England Research Environment (https://www.genomicsengland.co.uk/understanding-genomics/data). The 100,000 Genomes Project publication policies can be obtained from https://www.genomicsengland.co.uk/about-gecip/publications. Samples and results used in this study are provided in Genomics England under /re_gecip/shared_allGeCIPs/pancancer_drivers/results/. A MAF-like file detailing coding mutations across all 100kGP tumors analysed is available at /re_gecip/shared_allGeCIPs/pancancer_drivers/results/. The COSMIC and OncoKB clinical actionability data are available from https://cancer.sanger.ac.uk/actionability and https://www.oncokb.org/actionableGenes#sections=Tx, respectively. The canSAR chemogenomics data are available from https://cansar.ai/. The NHS Genomic Test Directory for Cancer is available from https://www.england.nhs.uk/publication/national-genomic-test-directories/. Lists of drivers from previous studies were obtained from COSMIC (https://cancer.sanger.ac.uk/cmc/home), IntOGen (https://www.intogen.org/search) and the The Cancer Genome Atlas (TCGA) Program pan-cancer analysis reported by ref. 13. Somatic mutations were annotated to the cached version of GRCh38 in VEP v.101.

## Code availability

Details and code for using the IntOGen framework are available at https://intogen.readthedocs.io/en/latest/index.html. The specific code to perform this analysis is available in the Genomics England research environment (https://re-docs.genomicsengland.co.uk/access/) under /re_gecip/shared_allGeCIPs/pancancer_drivers/code/. The link to becoming a member of the Genomics England research network and obtaining access can be found at https://www.genomicsengland.co.uk/research/academic/join-gecip. The code to perform the canSAR chemogenomics analysis is available through Zenodo (https://doi.org/10.5281/zenodo.8329054) (ref. 68).

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

## Acknowledgements

Funding was provided by the Wellcome Trust (214388), Cancer Research UK (C1298/A8362) and the Medical Research Council. A.S. is in receipt of a National Institute for Health Research (NIHR) Academic Clinical Lectureship, funding from the Royal Marsden Biomedical Research Centre, a starter grant for clinical lecturers from the Academy of Medical Sciences and a Wellcome Trust Early Career Award (227000/Z/23/Z). This is a summary of independent research supported by the NIHR Biomedical Research Centre at the Royal Marsden NHS Foundation Trust and the Institute of Cancer Research. This research was made possible through access to the data and findings generated by the 100,000 Genomes Project. The 100,000 Genomes Project is managed by Genomics England Limited (a wholly owned company of the Department of Health and Social Care). The 100,000 Genomes Project is funded by the National Institute for Health Research and NHS England. The Wellcome Trust, Cancer Research UK and the Medical Research Council also funded research infrastructure. The 100,000 Genomes Project uses data provided by patients and collected by the NHS as part of their care and support. We thank Genomics England for the communication regarding the sensitivity of WGS for detection of well-established cancer driver mutations.

## Author contributions

B.K., A.S. and R.H. designed the study. B.K., A.S., A.J.C. and D.C. performed sample curation. B.K., A.S., A.E., A.J.C., D.C., R.C., A.J.G., A.L., C.M. and D.W. performed bioinformatic and statistical analysis. B.K., A.S., A.E. and R.H. drafted the manuscript; all authors reviewed, read and approved the final manuscript.

## Competing interests

The authors declare no competing interests.

## Additional information

**Extended data** is available for this paper at https://doi.org/10.1038/s41588-024-01785-9.

**Correspondence and requests for materials** should be addressed to Richard Houlston.

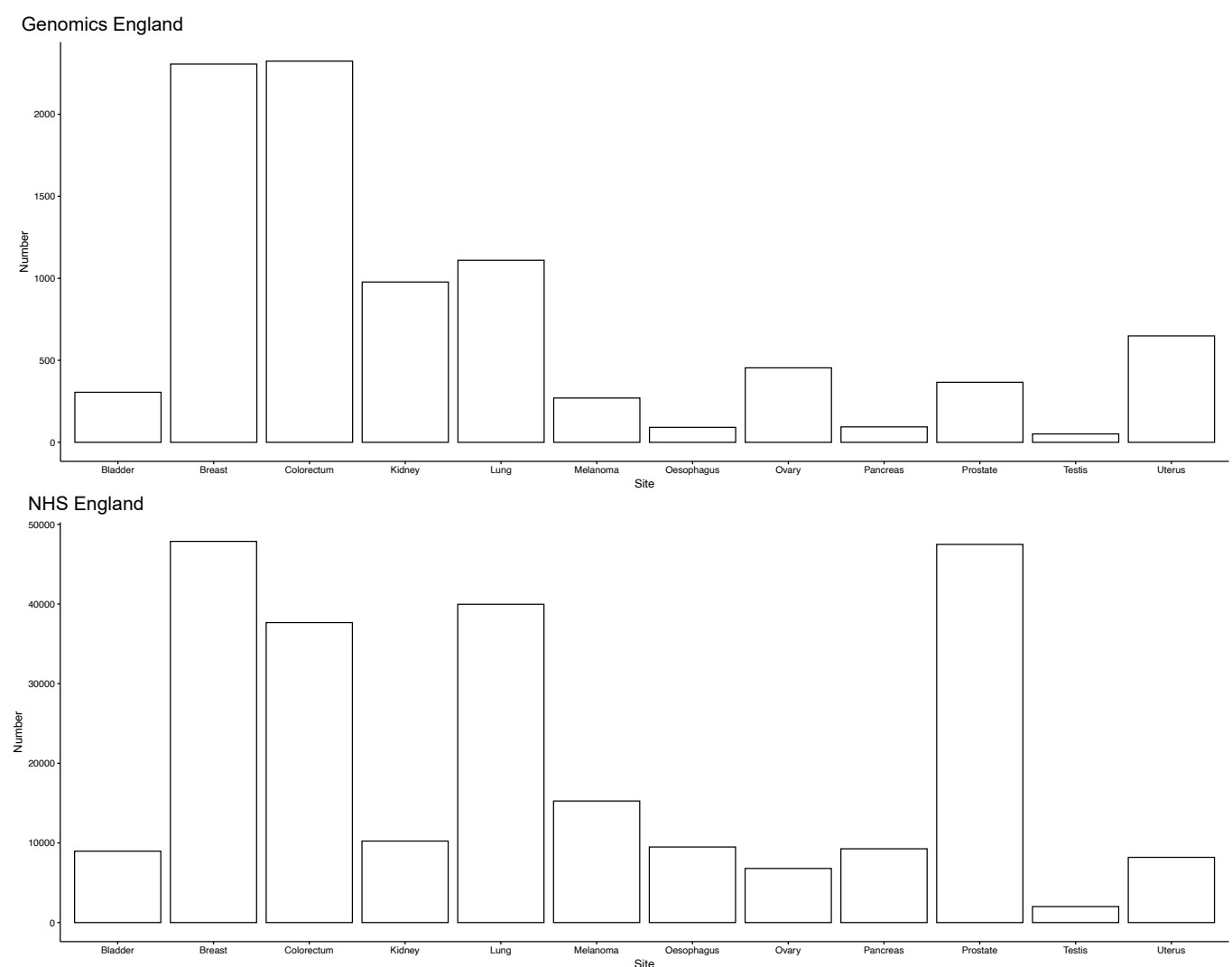

**Extended Data Fig. 1 | Comparison of number of samples per tumour type in the pan-cancer cohort compared to all cancer diagnosed in England in 2019.** Upper panel: the 100kGP cohort; lower panel: incidence of the different cancer types reported in the general population.

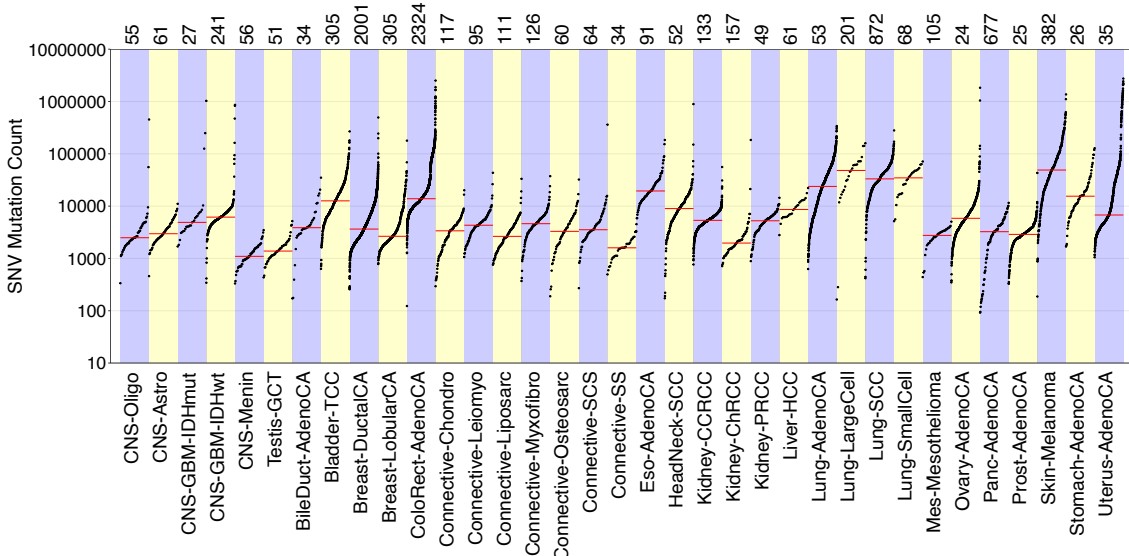

**Extended Data Fig. 2 | Mutation burden of tumours by each tumour type. The number of samples contributing to each tumour type are shown above the plot.** SNV, single nucleotide variant.

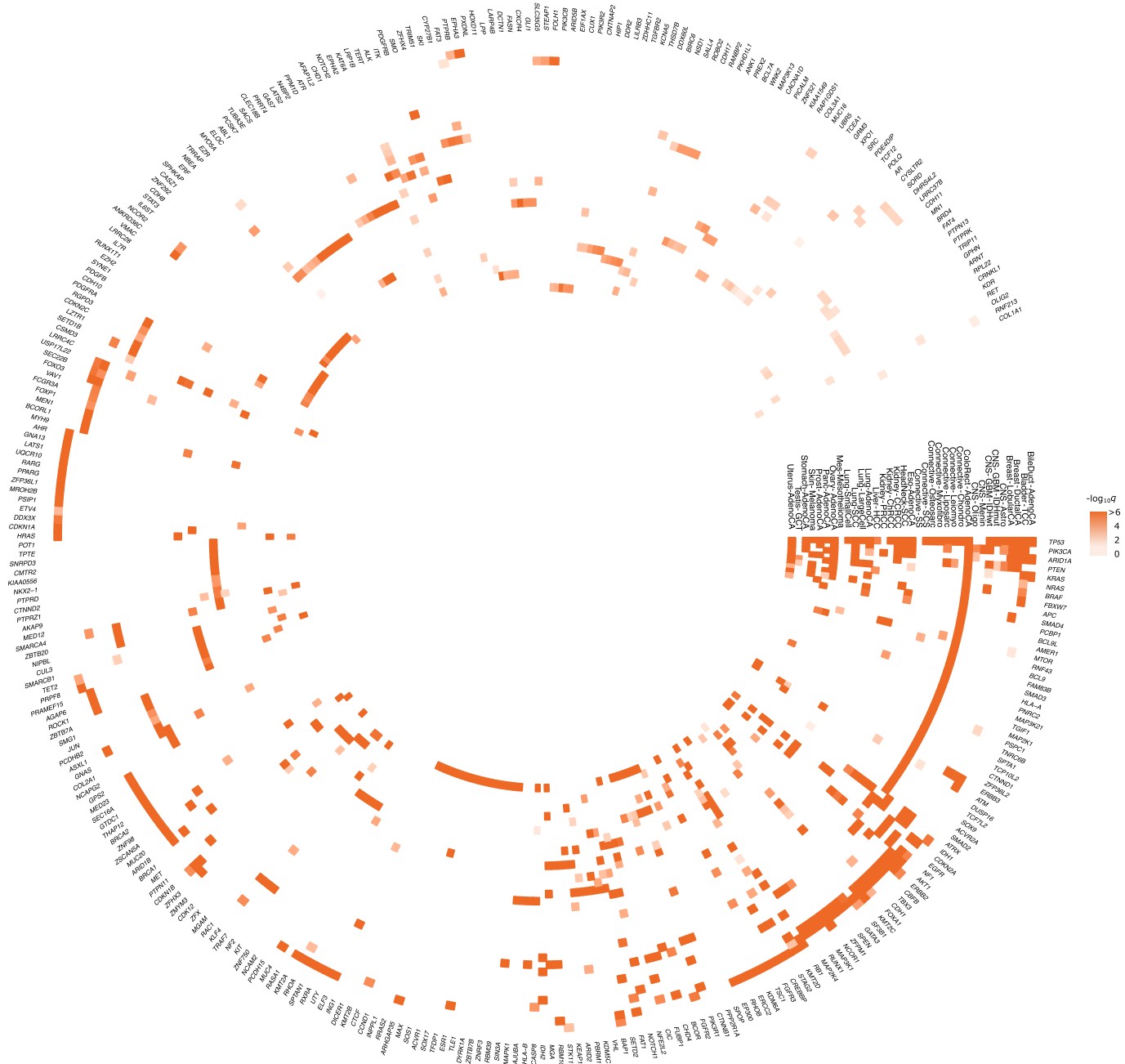

**Extended Data Fig. 3 | Circos heatmap of candidate cancer driver genes identified.** Heatmap intensity proportional to the *q* value.

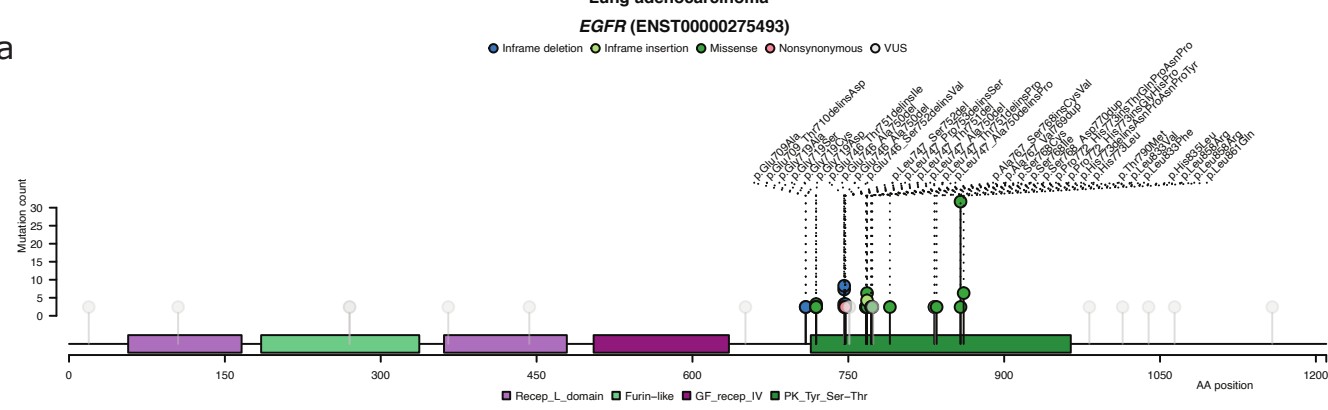

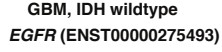

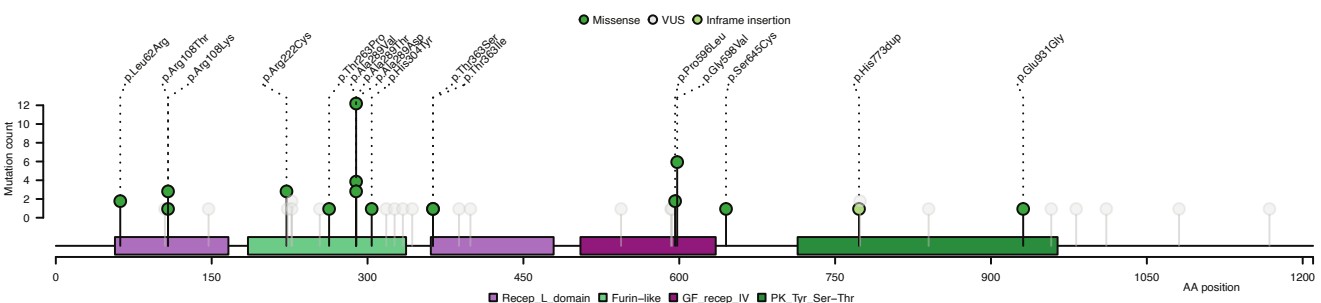

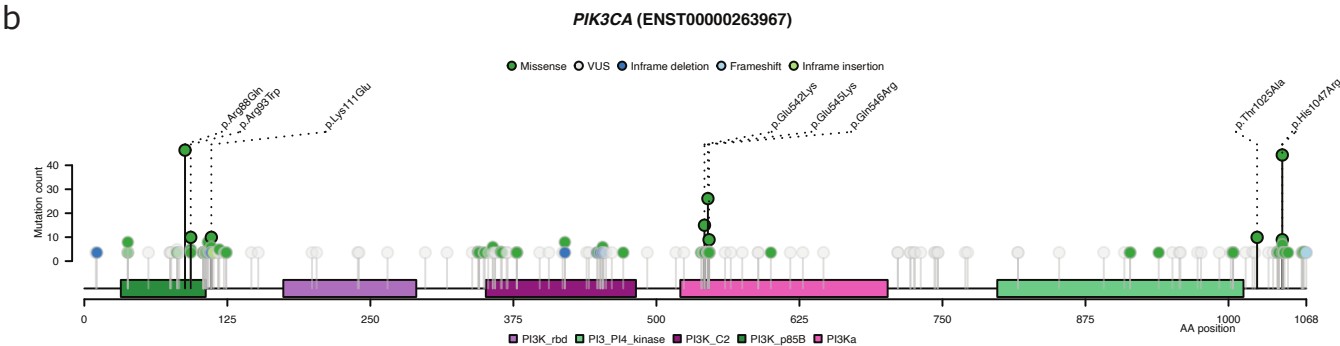

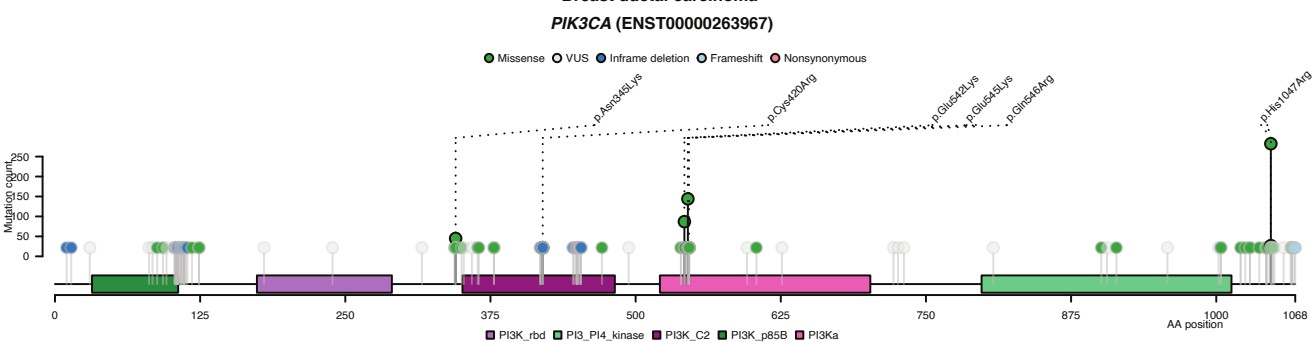

**Extended Data Fig. 4 | Mutation plots and pfam domain overlap for: (a)** *EGFR* **mutations in lung adenocarcinoma and GBM** *IDH* **wildtype; (b)** *PIK3CA* **mutations in uterine adenocarcinoma and breast ductal carcinoma.** Domain specific mutations were assessed by considering the cancer drivers where smRegions is a significant bidder (Q-value < 0.1) and the driver is annotated in multiple cancer types.

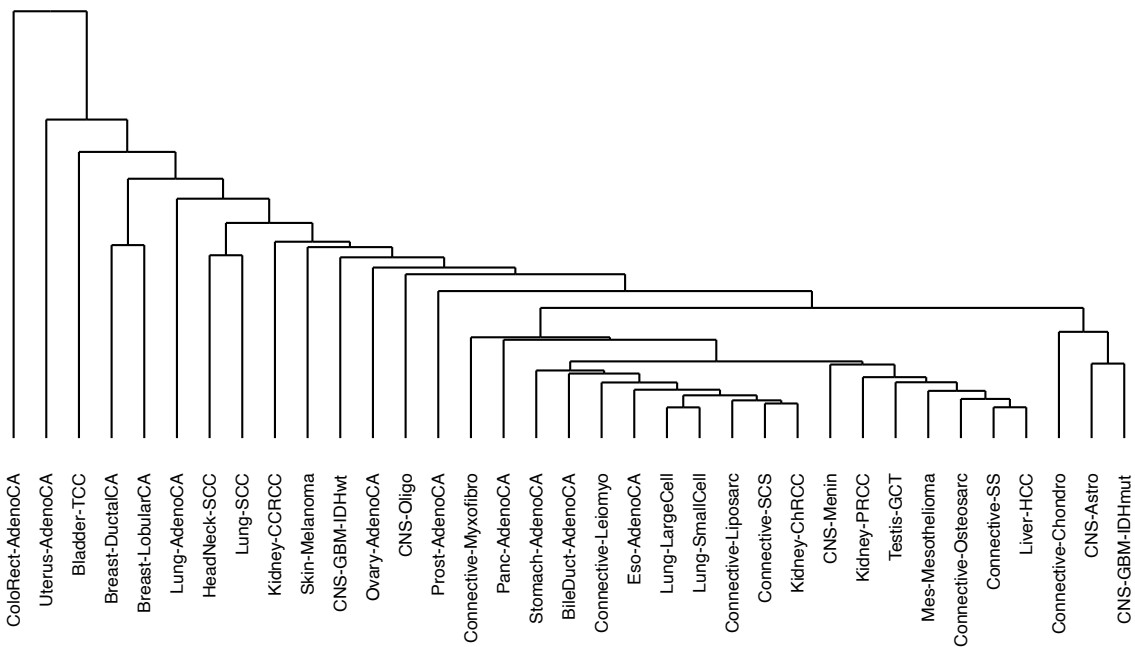

**Extended Data Fig. 5 | Hierarchical clustering of tumour types based on *P*-value of candidate driver genes across the 35 different tumour types.** Clustering performed using the hclust function in R.

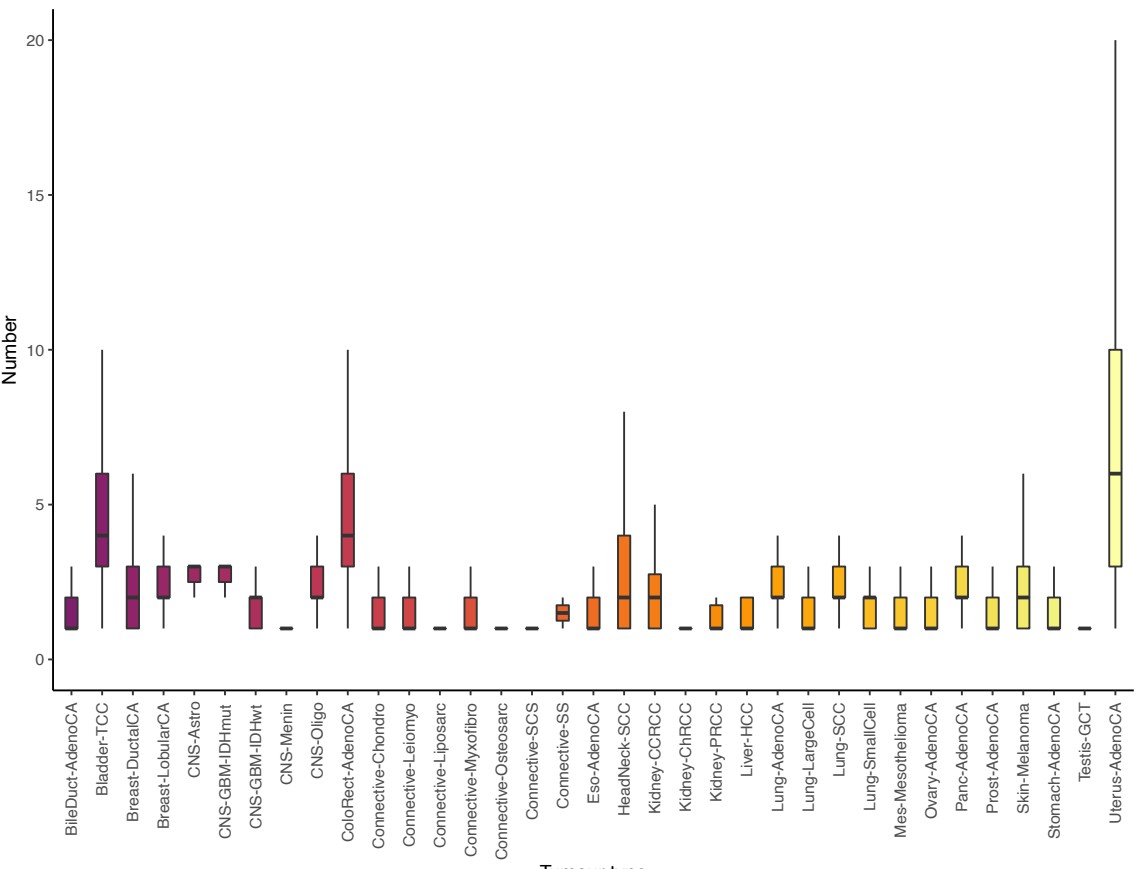

**Extended Data Fig. 6 | Per-tumour distribution of oncogenic mutations in tumour specific candidate cancer driver genes, across the 35 cancer types.** Analysis restricted to driver genes as predicted by IntOGen in the given cancer type. Oncogenicity predicted using OncoKB. The line within the box shows the median number of oncogenic mutations per sample in the cancer type. The box represents the interquartile range and whiskers represent the range.

# PANCANCER (Oncogenic)

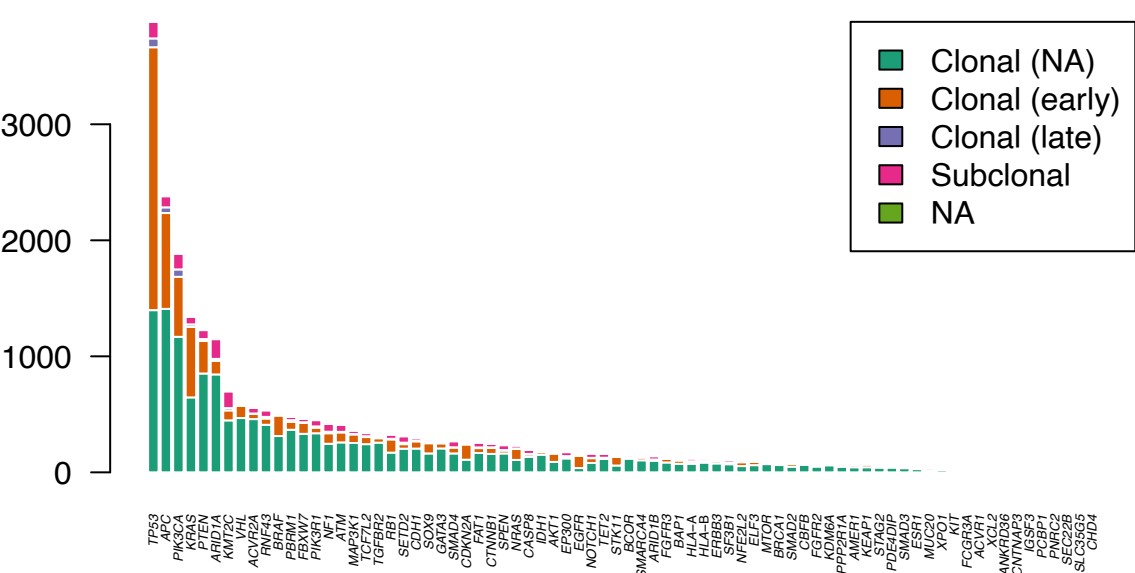

**Extended Data Fig. 7 | Oncogenic clonal and subclonal mutations across candidate driver genes across all tumor types.** Oncogenic clonal and subclonal mutations across candidate driver genes pan-cancer.

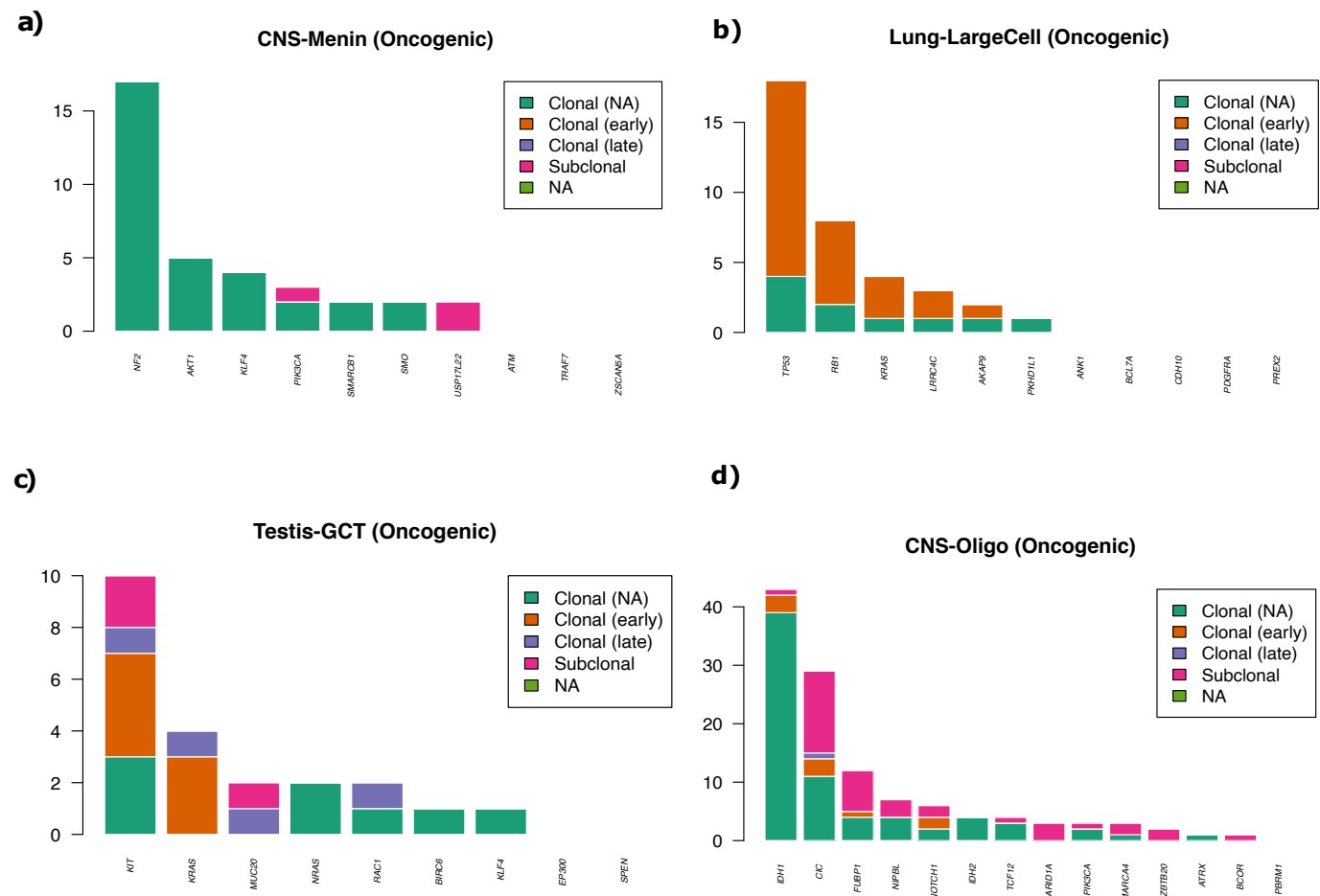

**Extended Data Fig. 8 | Oncogenic clonal and subclonal mutations across candidate driver genes.** Oncogenic clonal and subclonal mutations across candidate driver genes in: **a**) Meningioma; **b**) Large cell lung cancer; **c**) Testicular germ cell tumour; **d**) Oligodendroglioma.

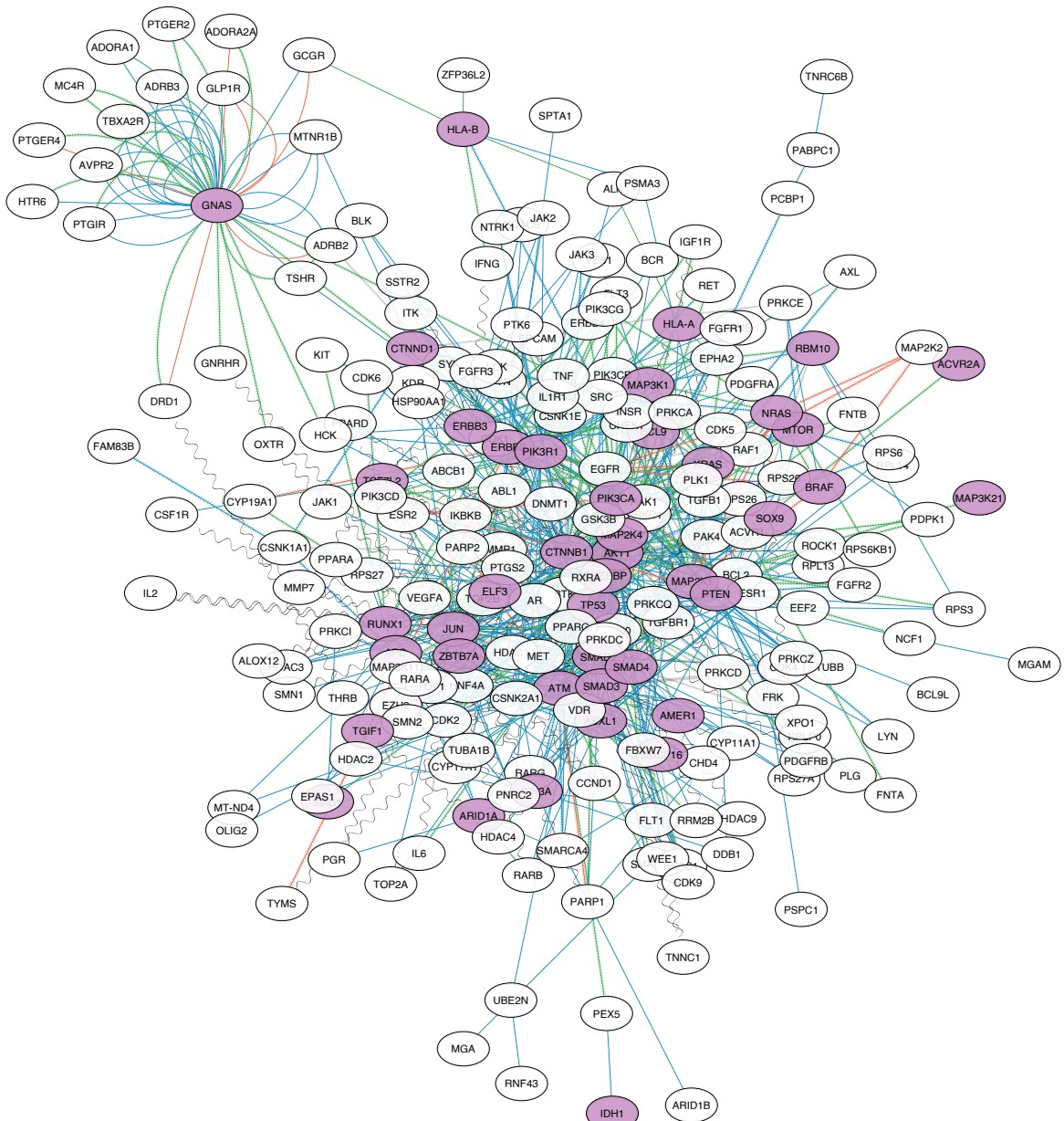

**Extended Data Fig. 9 | Example druggability network for colorectal cancer.**
Nodes acting as cancer-specific drivers are shaded purple. Edge visual properties
are as follows: OncoKB interactions, red contiguous arrow; Signor interactions,
green contiguous arrow; Signor inhibitors, black vertical slash; complex, black
zigzag; direct interaction, red solid line; direct X-ray interaction, green solid
line; direct non-protein data bank interaction, blue solid line; reaction, blue
contiguous arrow; transcriptional interaction, black sinewave. Figure generated
using Cytoscape[69].

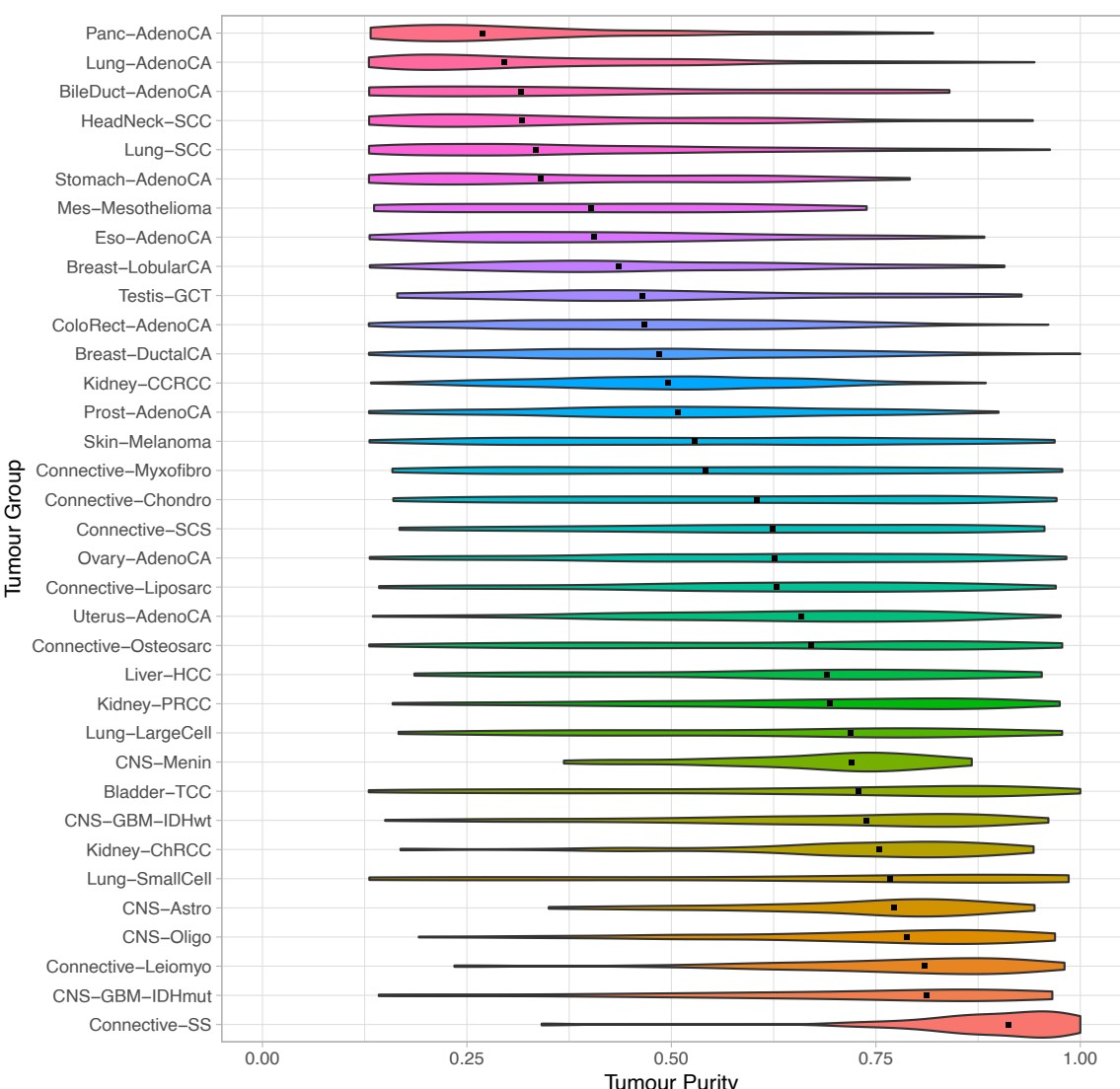

**Extended Data Fig. 10 | Violin plot of estimated tumour purity per cancer type.** Black square within each violin corresponds to the median value. Violin trimmed to the lowest and highest tumour purity estimate per cancer group. Purity estimates from Battenberg or Ccube.

# Reporting Summary

## Statistics

For all statistical analyses, confirm that the following items are present in the figure legend, table legend, main text, or Methods section.

| n/a | Confirmed | |
|---|---|---|
| ☐ | ☒ | The exact sample size (*n*) for each experimental group/condition, given as a discrete number and unit of measurement |
| ☐ | ☒ | A statement on whether measurements were taken from distinct samples or whether the same sample was measured repeatedly |
| ☐ | ☒ | The statistical test(s) used AND whether they are one- or two-sided<br>*Only common tests should be described solely by name; describe more complex techniques in the Methods section.* |
| ☐ | ☒ | A description of all covariates tested |
| ☐ | ☒ | A description of any assumptions or corrections, such as tests of normality and adjustment for multiple comparisons |
| ☐ | ☒ | A full description of the statistical parameters including central tendency (e.g. means) or other basic estimates (e.g. regression coefficient) AND variation (e.g. standard deviation) or associated estimates of uncertainty (e.g. confidence intervals) |
| ☐ | ☒ | For null hypothesis testing, the test statistic (e.g. *F*, *t*, *r*) with confidence intervals, effect sizes, degrees of freedom and *P* value noted<br>*Give P values as exact values whenever suitable.* |
| ☒ | ☐ | For Bayesian analysis, information on the choice of priors and Markov chain Monte Carlo settings |
| ☒ | ☐ | For hierarchical and complex designs, identification of the appropriate level for tests and full reporting of outcomes |
| ☐ | ☒ | Estimates of effect sizes (e.g. Cohen's *d*, Pearson's *r*), indicating how they were calculated |

*Our web collection on statistics for biologists contains articles on many of the points above.*

## Software and code

Policy information about availability of computer code

| Data collection | Samples were collected and processed by Genomics England. The code used for curation of samples is available inside the Genomics England Research Environment under /re_gecip/shared_allGeCIPs/pancancer_signatures/code/processClinicalData. |
|---|---|
| Data analysis | Details and code for using the Intogen framework are available here (https://intogen.readthedocs.io/en/latest/index.html). The specific code to perform this analysis is available in the Genomics England research environment under /re_gecip/shared_allGeCIPs/pancancer_drivers/ code/. The link to becoming a member of the Genomics England research network and obtaining access can be found here https:// www.genomicsengland.co.uk/research/academic/join-gecip. The code to perform the canSAR chemogenomics analysis is available through Zenodo (https://zenodo.org/record/8329054).<br>Additional packages/softwware used:<br>VerifyBamID v1.1.3 = https://github.com/statgen/verifyBamID<br>Ccube  v1 = https://github.com/keyuan/ccube<br>Isaac aligner v03.16.02.19 = https://github.com/Illumina/Isaac3<br>Strelka v2.4.7 = https://github.com/Illumina/strelka<br>bcftools v1.9 = https://samtools.github.io/bcftools/bcftools.html<br>alleleCount-FixVAF v4.1.0 = https://github.com/danchubb/alleleCount-FixVAF<br>VEP v101 = https://github.com/Ensembl/ensembl-vep<br>CADD v1.6 = https://github.com/kircherlab/CADD-scripts/<br>OncoKb v3.11 = https://www.oncokb.org/api-access<br>trackViewer v1.38.2 = https://github.com/jianhong/trackViewer<br>mSINGS = https://bitbucket.org/uwlabmed/msings/src/master/<br>HRDetect = https://github.com/eyzhao/hrdetect-pipeline |

```
Battenberg v2.2.8 =https://github.com/Wedge-lab/battenberg
Delly v0.7.9= https://github.com/dellytools/delly
Lumpy v0.2.13 = https://github.com/arq5x/lumpy-sv/releases
Manta v1.5.0 = https://github.com/Illumina/manta
GATK v.4.4.0 = https://github.com/broadinstitute/gatk
BEDOPS v2.4.2 = https://github.com/bedops/bedops
bedtools v2.3.0 = https://bedtools.readthedocs.io/en/latest/index.html
MutationTimeR v0.99.2  = https://github.com/gerstung-lab/MutationTimeR
```

For manuscripts utilizing custom algorithms or software that are central to the research but not yet described in published literature, software must be made available to editors and reviewers. We strongly encourage code deposition in a community repository (e.g. GitHub). See the Nature Portfolio guidelines for submitting code & software for further information.

# Data

Policy information about availability of data

All manuscripts must include a data availability statement. This statement should provide the following information, where applicable:
- Accession codes, unique identifiers, or web links for publicly available datasets
- A description of any restrictions on data availability
- For clinical datasets or third party data, please ensure that the statement adheres to our policy

Summary statistics for each tumour group are provided in the supplementary tables where such data does not enable identification of participants. All sample-specific WGS data and processed files from the 100,000 Genomes Project can be accessed by joining the Pan Cancer Genomics England Clinical Interpretation Partnership (GeCIP) Domain once an individual's data access has been approved (https://www.genomicsengland.co.uk/research/pan-cancer). The link to becoming a member of the genomics england research network and having access can be found here https://www.genomicsengland.co.uk/research/academic/join-gecip. The process involves an online application, verification by the applicant's institution, completion of a short information governance training course, and verification of approval by Genomics England. Please see https://www.genomicsengland.co.uk/research/academic for more information. The Genomics England data access agreement can be obtained from https://figshare.com/articles/dataset/GenomicEnglandProtocol_pdf/4530893/7. All analysis of Genomics England data must take place within the Genomics England Research Environment (https://www.genomicsengland.co.uk/understanding-genomics/data). The 100,000 Genomes Project publication policies can be obtained from https://www. genomicsengland.co.uk/about-gecip/publications. Samples and results used in this study are provided in Genomics England under /re_gecip/shared_allGeCIPs/pancancer_drivers/results/. A MAF-like file detailing coding mutations across all 100kGP tumours analysed is available at /re_gecip/shared_allGeCIPs/pancancer_drivers/results/. The COSMIC and OncoKB clinical actionability data are available from https://cancer.sanger.ac.uk/actionability and https://www.oncokb.org/actionableGenes#sections=Tx, respectively. The canSAR chemogenomics data are available from https://cansar.ai/. The NHS Genomic Test Directory for Cancer is available from https://www.england.nhs.uk/publication/national-genomic-test-directories/. List of drivers from prior studies obtained from COSMIC (https://cancer.sanger.ac.uk/cmc/home), Intogen (https://www.intogen.org/search) and and the The Cancer Genome Atlas (TCGA) Program pan-cancer analysis reported by Bailey et al. Somatic mutations were annotated to the cached version of GRCh38 in VEP v101.

# Research involving human participants, their data, or biological material

Policy information about studies with human participants or human data. See also policy information about sex, gender (identity/presentation), and sexual orientation and race, ethnicity and racism.

| | |
|---|---|
| Reporting on sex and gender | Sex was used as reported by NHSD, PHE/NCRAS and the GMCs where this matched the inferred sex from genomic sequencing. Where they do not match the sample was excluded. |
| Reporting on race, ethnicity, or other socially relevant groupings | Reported race, ethnicity, or other socially relevant groupings were not used in this study. |
| Population characteristics | Information relating to the cohort in this analysis are provided in supplementary table 3. The collection and processing of treatment information is described in detail in the methods. |
| Recruitment | Clinical and demographic data were obtained from NHS Digital (NHSD), Public Health England's National Cancer Registration and Analysis Service (PHE-NCRAS) and the Genomic Medicine Centres (GMCs) through the Genomics England Research Environment. |
| Ethics oversight | The 100,000 Genomes Project protocol was approved by the East of England and South Cambridge Research Ethics Committee on 20 February 2015, REC reference 14/EE/1112 |

Note that full information on the approval of the study protocol must also be provided in the manuscript.

# Field-specific reporting

Please select the one below that is the best fit for your research. If you are not sure, read the appropriate sections before making your selection.

☒ Life sciences    ☐ Behavioural & social sciences    ☐ Ecological, evolutionary & environmental sciences

For a reference copy of the document with all sections, see nature.com/documents/nr-reporting-summary-flat.pdf

# Life sciences study design

All studies must disclose on these points even when the disclosure is negative.

| | |
|---|---|
| Sample size | 10,478 samples were included in the full cohort. Exact sample sizes for tumour groups are provided in supplementary table 2. Sample size was chosen based on the availability of whole genome sequencing of tumour/normal pairs in the Genomics England research environment. |
| Data exclusions | A detailed description of the sample quality control is provided in the methods. Supplementary table 1 provides information on how many samples were excluded. Sequenced tumour samples were excluded if clinical data were missing or if unresolvable conflicts existed between the clinical data sources (GMCs, NHSD, PHE-NCRAS, histology reports). In total 2,251/14,129 (15.9%) of tumour samples were excluded based on availability and consistency of reported sex, tumour histology, tumour type, sampling date or if the participant was recorded as less than 18 years old at the time of sampling. 267/11878 (2.2%) of tumour samples with required clinical data available were excluded based on tumour sample purity and sequencing data quality. Duplicate tumour samples were also removed, to ensure that no individual was represented more than once in a tumour group. If multiple sequenced tumour samples from the same tumour group were available for an individual, we preferentially kept primary tumour samples with highest purity. Non-solid tumours were removed from this analysis. Based on these criteria, 10,478 tumour samples were suitable for analysis. |
| Replication | This study has an observational rather than an experimental study design, and only one sample was sequenced from each participant, in the great majority of cases. We replicate many of the findings from previously published studies of somatic cancer driver genes. |
| Randomization | This study has an observational rather than an experimental study design hence randomisation of study participants is not relevant. |
| Blinding | This study used real-world observation data collected from NHS trusts. The investigators did not have control over sample selection, collection and processing and as such blinding is not relevant to this study. |

# Reporting for specific materials, systems and methods

We require information from authors about some types of materials, experimental systems and methods used in many studies. Here, indicate whether each material, system or method listed is relevant to your study. If you are not sure if a list item applies to your research, read the appropriate section before selecting a response.

## Materials & experimental systems

| n/a | Involved in the study |
|---|---|
| ☒ | ☐ Antibodies |
| ☒ | ☐ Eukaryotic cell lines |
| ☒ | ☐ Palaeontology and archaeology |
| ☒ | ☐ Animals and other organisms |
| ☒ | ☐ Clinical data |
| ☒ | ☐ Dual use research of concern |
| ☒ | ☐ Plants |

## Methods

| n/a | Involved in the study |
|---|---|
| ☒ | ☐ ChIP-seq |
| ☒ | ☐ Flow cytometry |
| ☒ | ☐ MRI-based neuroimaging |

## Plants

| | |
|---|---|
| Seed stocks | *Report on the source of all seed stocks or other plant material used. If applicable, state the seed stock centre and catalogue number. If plant specimens were collected from the field, describe the collection location, date and sampling procedures.* |
| Novel plant genotypes | *Describe the methods by which all novel plant genotypes were produced. This includes those generated by transgenic approaches, gene editing, chemical/radiation-based mutagenesis and hybridization. For transgenic lines, describe the transformation method, the number of independent lines analyzed and the generation upon which experiments were performed. For gene-edited lines, describe the editor used, the endogenous sequence targeted for editing, the targeting guide RNA sequence (if applicable) and how the editor was applied.* |
| Authentication | *Describe any authentication procedures for each seed stock used or novel genotype generated. Describe any experiments used to assess the effect of a mutation and, where applicable, how potential secondary effects (e.g. second site T-DNA insertions, mosiacism, off-target gene editing) were examined.* |

