## [Peer Review File · Nature Genetics]

Peer Review Information

Manuscript Title: Analysis of 10,478 cancer genomes identifies candidate driver genes and opportunities for precision oncology

Corresponding author name(s): Professor Richard Houlston

Editorial Notes:

Transferred manuscripts This document only contains reviewer comments, rebuttal and decision letters for versions considered at Nature Genetics.

Reviewer Comments & Decisions:

Decision Letter, initial version:

20th Nov 2023

Dear Professor Houlston,

Your Article entitled "Cancer driver genes and opportunities for precision oncology revealed by whole genome sequencing 10,478 cancers" has now been seen by 2 referees (unfortunately, Reviewer #3 was unable to re-review), whose comments are attached. While they find your work of potential interest, they have raised serious concerns which in our view are sufficiently important that they preclude publication of the work in Nature Genetics, at least in its present form.

While the referees find your work of some interest, they raise concerns about the strength of the novel conclusions that can be drawn at this stage.

Should further experimental data allow you to fully address these criticisms we would be willing to consider an appeal of our decision (unless, of course, something similar has by then been accepted at Nature Genetics or appeared elsewhere). This includes submission or publication of a portion of this work someplace else.

The required new experiments and data include, but are not limited to those detailed here. We hope you understand that until we have read the revised manuscript in its entirety we cannot promise that it will be sent back for peer review.

If you are interested in attempting to revise this manuscript for submission to Nature Genetics in the

future, please contact me to discuss a potential appeal. Otherwise, we hope that you find our referees' comments helpful when preparing your manuscript for resubmission elsewhere.

Although we cannot publish your paper, it may be appropriate for another journal in the Nature Portfolio. If you wish to explore the journals and transfer your manuscript please use our manuscript transfer portal. You will not have to re-supply manuscript metadata and files, unless you wish to make modifications. For more information, please see our manuscript transfer FAQ page.

Sincerely,

Safia Danovi
Editor
Nature Genetics

Reviewers' Comments:

Reviewer #1:

Remarks to the Author:

Summary:

A re-review of this manuscript was very difficult without a track changes file and with very brief responses to reviewers which did not include exact changes in text. There was clearly a lot of work put into this manuscript, and the 100,000 genomes are a tremendous resource, but this work is still not detailed (gene level vs variant level tables) or reproducible.

Major comments:

- Supplementary Table 5 should contain variant-level annotation.
- On reviewer 2's comment about 75% failure of oncology drugs – this is a misinterpretation of [https://www.cell.com/trends/molecular-medicine/fulltext/S1471-4914\(01\)01986-4?roistat_visit=14074330](https://www.cell.com/trends/molecular-medicine/fulltext/S1471-4914(01)01986-4?roistat_visit=14074330) by the authors. Table 1 in that manuscript shows that 25% of chemotherapy (alone) fails in cancer. I agree with reviewer 2 – the authors are extrapolating to all drugs and this is incorrect and the manuscript still contains this statement. It should be removed.
- Similar to my first review, the data and code are insufficient. While the authors state: "We are limited in providing identifiable patient information, as determined by Genomics England, in any publication. We have however now included details of a repository in the secure Genomics England research environment where such data can be found. The code used in this analysis will be made available via GitHub and Zenodo."
 - o It is common practice to release somatic variant calls and genomic summary results in manuscripts such as this and that does not compromise patient identifiability. Can pseudo IDs be re-generated or variant-level details provided without any IDs? The variant details are critical to enable others to further this research and really need to be included in this type of manuscript. Supplemental Table 10 – "Oncogenic mutations" – these need to be described.
 - o The reviewers cannot access Genomics England if not a member. There is no way for this reviewer to reproduce any of the analyses to confirm the results. There is one minor analysis code on Zenodo, but nothing yet on GitHub which is publicly available.
- Figure 2a in no way will be legible as a panel in a 4-figure panel, as it is not legible now. Additionally, it is not easy for a person to go around in a circle to read- horizontal reading is ideal. It

would help if the authors put the figures into panels for the reviewers.

Reviewer #2:

Remarks to the Author:

In reviewing this a second time, many of the revisions are improvements but there still are some stumbling blocks. The paper has great potential and needs to be clear in its intent content and context- it is this latter issue that the brevity of discussion challenges. If this is to be published, the authors should speak more to the issues at hand so the reader can understand and use the knowledge for next steps.

The discussion remains very curt and does not fully address the issues of selection bias (or lack thereof here) and its implications- nor does it address coverage and its implications for detection sensitivity/specificity. It is also unusual to not discuss mRNA and methylation data- which can be informative for targeted therapy- such as the fusion genes and methylation-related therapies not covered (understandably here).

For instance, if this paper, which is a Landscape, wide-sweeping assessment is to be useful and read wisely, then the context needs to be carefully presented. For instance, the authors have dodged the previous comment about the failure rates which is substantively overstated- perhaps for overly dramatic purposes and continue to use two very unusual references- one from 2001 and the other from 2018 that is specific to a small fraction of what has been accumulated in this collection of available tumors. Please, be more realistic and accurate in portraying why this is informative and use the widely available data from the UK and the US- which do NOT support the overly cynical 75% failure of standard therapies....BTW- how do we get to 62-64% survival overall? Second line therapies are not as effective as primary....in their response they indicate targeted therapy but the text does not portray this- 'standardized therapy' is NOT equivalent to targeted therapy.

The abstract raises the question of perhaps 96 additional targets, to date, but the discussion of this lacks a deeper and informative understanding of next steps. The precision of estimates is naïve and overstated- as is the contention that one can make a precise statement when in fact, there are so many factors/biases inherent in this study- ones that should not be viewed with such absolutism. In this regard, it is misguided and misleading. The numbers are 'estimates' as clearly implied for additional theoretical studies. Counting instances and viewing as a precise number is only so useful if there is a message going forward...

On the positive side, they have added very informative analyses on timing clonality and co-occurrence. The discussion has been augmented adequately (knowing space is an issue- a shame here) and so the requested minimal issues of the 'epidemiology of the study are discussed' is addressed. One small point is the bias that only samples with large freshfrozen material that meet a certain criteria could be included- something that needs a word or two. Tumor size has meaning and implications for metastases and outcomes.

This reviewer would like a comment about coverage of WGS as this is a major economic issue that could be invoked with concerning consequences. The brief discussion about targeted sequencing, usually of high coverage, should at minimum address WGS coverage.

This reviewer agrees with the author's response to some of fellow reviewer #3's comments on access- this is a data set that can be reached but its process is arduous and the ability compute over the data set inconvenient but not impossible. The authors are accurate and this reviewer wishes that this discussion could lead to a more useful accession policy.....

Decision Letter, Appeal:

5th Feb 2024

Dear Dr Houlston,

Thank you for asking us to reconsider our decision on your manuscript "Novel cancer driver genes and opportunities for precision oncology revealed by analysis of 10,478 cancer genomes". I have now discussed the points of your letter with my colleagues, and we invite you to revise your manuscript along the lines that you propose.

When preparing a revision, please ensure that it fully complies with our editorial requirements for format and style; details can be found in the Guide to Authors on our website (<http://www.nature.com/ng/>).

Please be sure that your manuscript is accompanied by a separate letter detailing the changes you have made and your response to the points raised. At this stage we will need you to upload:

1) a copy of the manuscript in MS Word .docx format.

2) The Editorial Policy Checklist:

<https://www.nature.com/documents/nr-editorial-policy-checklist.pdf>

3) The Reporting Summary:

(Here you can read about the role of the Reporting Summary in reproducible science:

<https://www.nature.com/news/announcement-towards-greater-reproducibility-for-life-sciences-research-in-nature-1.22062>)

Please use the link below to be taken directly to the site and view and revise your manuscript:

[redacted]

With kind wishes,

Safia Danovi
Editor
Nature Genetics

Author Rebuttal to Initial comments

Reviewer #1:

- 1.1 A re-review of this manuscript was very difficult without a track changes file and with very brief responses to reviewers which did not include exact changes in text.

Response: We now provide a tracked change file, comparing the latest version of the manuscript with the first version.

- 1.2 There was clearly a lot of work put into this manuscript, and the 100,000 genomes are a tremendous resource, but this work is still not detailed (gene level vs variant level tables) or reproducible.

Response: We agree with the reviewer that access to data and code is critical in ensuring the work is reproducible and is of benefit to the wider scientific community. We are however restricted in our ability to export data out of the Genomics England Research Environment, because of the governance issues in place. We would like to draw attention to the fact that reviewer 2 is aware of the restrictions placed upon us from Genomics England regarding data accessibility. Other articles which utilise genetic or phenotypic data from Genomics England have not made such data available (e.g. <https://www.nature.com/articles/s41588-022-01211-y>, <https://genomemedicine.biomedcentral.com/articles/10.1186/s13073-022-01084-0>, <https://www.nature.com/articles/s41591-023-02211-z>, <https://www.science.org/doi/10.1126/science.abl9283>). Moreover, other large-scale genomic studies place similar restrictions regarding data dissemination.

Further information regarding data accessibility and Genomics England can be found here: <https://re-docs.genomicsengland.co.uk/airlock/>, but we will provide a summary below.

Genomics England participants have given consent for their data to be used as follows: ‘...although researchers can look at your data and ask questions about it, they can only take away the answers to their questions (their results). They can’t copy or take away any of your individual data’. For this reason, Genomics England does not permit the release of data which could lead to the re-identification of any participant outside the Research Environment – either from the material alone, or through aggregation with other data available now or in the future.

Our analysis was therefore conducted within the Research Environment (<https://www.genomicsengland.co.uk/research/research-environment>) and the only data we can export are analytical results. We have had extensive and detailed discussions with Genomics England as to what data we can provide with this manuscript. We have been explicitly forbidden by Genomics England to provide somatic variant-level information with pseudo-anonymised identifiers due to the risk that such data could identify individuals.

We would like to highlight that the genomic and clinical data and code associated with this manuscript is accessible in the Genomics England Research Environment (<https://www.genomicsengland.co.uk/research/research-environment>). This allows anyone to reproduce our analysis. To access genomic and clinical data, an application must be made to become a member of either the Genomics England Research Network or the Discovery Forum (industry partners) (<https://www.genomicsengland.co.uk/research/academic>). Membership is available worldwide to academic researchers as well as charities and government departments that carry out research.

The process for joining the Research Network consists of the following steps:

- Your institution will need to sign a participation agreement and email the signed version to gecip-help@genomicsengland.co.uk
- Choose a GECIP of interest and apply to join through the online form
- Genomics England will review your application within ten working days
- Your institution will validate your affiliation
- Following completion of online Information Governance training (30-60 minutes), you will be granted access to the Research Environment within two working days

Upon gaining access to the Research Environment, code and results files associated with the manuscript can be accessed in the following locations:

```
/re_gecip/shared_allGeCIPs/pancancer_drivers/code
```

```
/re_gecip/shared_allGeCIPs/pancancer_drivers/results
```

We hope the above demonstrates we have provided the maximum amount of data available as permitted by Genomics England.

1.3 Supplementary Table 5 should contain variant-level annotation.

Response: Please see the response to 1.2.

1.4 On reviewer 2's comment about 75% failure of oncology drugs – this is a misinterpretation of <https://www.cell.com/trends/molecular-medicine/fulltext/S1471-4914> [cell.com](01)01986-4?roistat_visit=14074330 by the authors. Table 1 in that manuscript shows that 25% of chemotherapy (alone) fails in cancer. I agree with reviewer 2 – the authors are extrapolating to all drugs and this is incorrect and the manuscript still contains this statement. It should be removed.

Response: We acknowledge this point and have revised our introduction extensively and removed commentary about drug failure:

“Precision oncology aims to tailor cancer therapy to the unique biology of the patient’s cancer thereby optimising treatment efficacy and minimising toxicity^{1,2}. Underpinning precision oncology is the concept of somatic driver mutations being the foundation of cancer biology^{3,4}.

The expansion in the number of therapeutically actionable genes has exposed the limitations of single-analyte genomic assays in cancer⁵. The modest incremental cost of adding additional cancer genes to high-throughput sequencing-based panels has made the development of drugs targeting increasingly smaller molecularly defined subsets of patients with cancer financially and logistically feasible⁶. The development of inhibitors effective in cancers driven by rare genomic mutations has required the concurrent development of novel clinical trial designs such as basket trials in which eligibility is based on mutational status instead of organ site⁷. With the advent of basket studies, many oncologists now consider that tumour genomic profiling should be offered to all patients with cancer who are not candidates for curative-intent local or systemic therapy⁸.

Currently, multiple standalone tests or a panel are typically used to capture a set of genomic features in a tumour to inform patient treatment⁹. However, falling costs are making whole genome sequencing (WGS) a potentially attractive proposition as a single all-encompassing test to identify cancer drivers and other genomic features, which may not be captured by standard testing but are clinically actionable¹⁰. This approach is being explored by the 100,000 Genomes

Project (100kGP), which is seeking to deliver the vision of precision oncology through WGS to National Health Service (NHS) patients as part of their routine care¹¹.

Herein we report an analysis of WGS data on 10,478 patients spanning 35 cancer types recruited to the 100kGP (**Fig. 1a**). Across all cancer types we identify 330 driver genes, including 74 which are novel to any cancer. We relate these to their actionability both in terms of currently approved therapeutic agents and through computational chemogenomic analysis to predict candidacy for future clinical trials.”

- 1.5 Similar to my first review, the data and code are insufficient. While the authors state: “We are limited in providing identifiable patient information, as determined by Genomics England, in any publication. We have however now included details of a repository in the secure Genomics England research environment where such data can be found. The code used in this analysis will be made available via GitHub and Zenodo.” It is common practice to release somatic variant calls and genomic summary results in manuscripts such as this and that does not compromise patient identifiability. Can pseudo IDs be re-generated or variant-level details provided without any IDs? The variant details are critical to enable others to further this research and really need to be included in this type of manuscript.

Response: Please see the response to 1.2 regarding data availability. With respect to the availability of code used in this analysis, code for using the IntOGen framework are available here (<https://intogen.readthedocs.io/en/latest/index.html>). The code to perform the canSAR chemogenomics analysis is available through Zenodo (<https://zenodo.org/record/8329054>). As we did not develop any novel code to perform the driver gene analysis, we believe the most helpful way to make the code available is in conjunction with the underlying data so as to ensure the findings are reproducible. Hence, as requested, this code is available in the Genomics England research environment at the following location: `/re_gecip/shared_allGeCIPs/pancancer_drivers/code/`.

- 1.6 **Supplemental Table 10 – “Oncogenic mutations” – these need to be described.**

Response: We have provided a definition of oncogenic mutations, as per OncoKB, in the legend of Supplementary Table 10.

- 1.7 The reviewers cannot access Genomics England if not a member. There is no way for this reviewer to reproduce any of the analyses to confirm the results. There is one minor analysis code on Zenodo, but nothing yet on GitHub which is publicly available.

Response: Please see the response to 1.2 and 1.5.

- 1.7 Figure 2a in no way will be legible as a panel in a 4-figure panel, as it is not legible now. Additionally, it is not easy for a person to go around in a circle to read- horizontal reading is ideal. It would help if the authors put the figures into panels for the reviewers.

Response: We have changed Figure 2A into a rectangular stand alone figure and present only drivers which are present in >2 cancer types as requested. However, we believe the full circos plot at high resolution is more informative to the reader, particularly when viewed online. We welcome advice from the editor regarding what is acceptable for an online publication. Pending any decision we now provide the original Figure as Supplementary Figure 3.

Reviewer #2:

- 2.1 In reviewing this a second time, many of the revisions are improvements but there still are some stumbling blocks. The paper has great potential and needs to be clear in its intent content and context- it is this latter issue that the brevity of discussion challenges. If this is to be published, the authors should speak more to the issues at hand so the reader can understand and use the knowledge for next steps.

Response: We thank the reviewer for acknowledging our manuscript is much improved. We have revised the manuscript to ensure that relevant issues are discussed as requested by the reviewers.

- 2.2 The discussion remains very curt and does not fully address the issues of selection bias (or lack thereof here) and its implications- nor does it address coverage and its implications for detection

sensitivity/specificity. It is also unusual to not discuss mRNA and methylation data- which can be informative for targeted therapy- such as the fusion genes and methylation-related therapies not covered (understandably here).

Response: We acknowledge this and the following paragraphs are now included in the discussion:

“The strengths of this study not only include the cohort size, but the combination of systematic processing of samples and data arising from multiple treatment centres across England. These strengths minimise the impact of between-centre sequencing effects while ensuring a representative cohort of cancers are captured⁴⁶. We do however, acknowledge that while the spectrum of cancers included in our analysis are largely representative of those diagnosed in the UK, patients recruited to 100kGP are younger and predominantly have early-stage disease. Furthermore, 92% of the patients had self-reported European ancestry. Since characteristics such as patient ancestry and geography can affect the mutagenic profile of tumours, this potentially impacts on the generalisability of our findings to worldwide populations^{47,48}.”

“Despite the merits of WGS as a one-stop clinical assay, its wider adoption outside selected academic and commercial centres has been limited⁴². A major hurdle is that the tumour material available for many patients is of insufficient quantity, quality or purity for these broader sequencing platforms. Indeed, in the 100kGP the lack of access to fresh frozen samples (and/or those of sufficient quantity) precluded the analysis of tumours from many patients¹¹. In designing clinical assays, the limitations imposed by cost and sequencing capacity require the balancing of sequencing breadth and depth⁴⁶. Presently, this trade-off, the higher coverage of targeted assays, represents an advantage over WGS for detection of alterations in genes clinically validated as biomarkers of drug response, especially in samples with poor DNA quality or high stromal contamination. A wider adoption of WGS will require further reductions in sequencing costs and technological improvements to enable the use of lower-quality, archival formalin-fixed, paraffin-embedded (FFPE) tumour tissue⁶⁰. Any such developments will have to address the issue that formalin-fixation adversely affects DNA quality and the ability to reliably call variants from WGS data, even when employing bioinformatic correction^{46,61,62}. Aside from such technical issues there are also inherent limitations to short-read WGS. Notably structural variants cannot be robustly called with low concordance being a feature of currently implemented algorithms^{63,64}. It is likely this limitation will only be addressed by adoption of long-read sequencing, albeit currently this incurs a high requirement for DNA and additional cost, thus restricting its use in the diagnostic setting⁶⁵. The continued decline in sequencing costs and the identification of new genomic biomarkers predictive of drug response has driven the rapid adoption of multigene profiling of

patients as a component of routine cancer care. As our analysis indicates, the future adoption of WGS or broader panels should enable more accurate assessments of the driver mutational landscape predictive of drug response.”

- 2.3 For instance, if this paper, which is a Landscape, wide-sweeping assessment is to be useful and read wisely, then the context needs to be carefully presented. For instance, the authors have dodged the previous comment about the failure rates which is substantively overstated- perhaps for overly dramatic purposes and continue to use two very unusual references- one from 2001 and the other from 2018 that is specific to a small fraction of what has been accumulated in this collection of available tumors. Please, be more realistic and accurate in portraying why this is informative and use the widely available data from the UK and the US- which do NOT support the overly cynical 75% failure of standard therapies....BTW- how do we get to 62-64% survival overall? Second line therapies are not as effective as primary.... in their response they indicate targeted therapy but the text does not portray this- ‘standardized therapy’ is NOT equivalent to targeted therapy.

Response: Please see the response to 1.4.

- 2.4 The abstract raises the question of perhaps 96 additional targets, to date, but the discussion of this lacks a deeper and informative understanding of next steps. The precision of estimates is naïve and overstated- as is the contention that one can make a precise statement when in fact, there are so many factors/biases inherent in this study- ones that should not be viewed with such absolutism. In this regard, it is misguided and misleading. The numbers are ‘estimates’ as clearly implied for additional theoretical studies. Counting instances and viewing as a precise number is only so useful if there is a message going forward...

Response: We acknowledge this point and have removed the precise estimate of the number of additional targets from our abstract. Furthermore, we have included an additional discussion in the paragraph regarding the value of WGS:

“A barrier to the broader success of precision oncology paradigms may be the large number of ‘undruggable’ oncogenic mutations coupled with the fact that targeting downstream effectors typically fail to demonstrate the levels of clinical efficacy of drugs that directly inhibit the mutated

oncogene. Recent developments in protein structure prediction, novel degraders, covalent inhibition and allosteric protein domain maps seek to unlock these ‘undruggable’ proteins^{51–54}. Furthermore, WGS allows for the extension of analyses beyond the consideration of individual genetic alterations thereby affording a clinically significant benefit over targeted panel sequencing assays. Mutational signatures associated with dMMR and HRD are increasingly being shown to be clinically relevant to defining responsiveness to immunotherapy and PARP inhibition respectively^{33,38}. Additionally, there is increasing evidence that other signatures reflecting the DNA repair capacity of cancer cells predictive of drug responsiveness to other agents are being recognised^{5,55}. A more detailed discussion and comprehensive description of all classes of mutational signatures observed across the 100kGP are reported in our companion paper¹⁸. The ability to robustly characterise mutational signatures may therefore prove to be a major clinically significant incremental benefit of WGS over targeted panel sequencing assays. Moreover, the provision of WGS is likely to play a greater role in patient management given T-cell based therapies are of increasing importance and *in silico* approaches are now used to predict the presence of immunogenic tumour-specific neoantigens from WGS^{56–59}.”

- 2.5 On the positive side, they have added very informative analyses on timing clonality and co-occurrence. The discussion has been augmented adequately (knowing space is an issue- a shame here) and so is the requested minimal issues of the ‘epidemiology of the study are discussed’ is addressed.

Response: We thank the reviewer for acknowledging the additional informative analysis of epidemiology, timing clonality and co-occurrence.

- 2.6 One small point is the bias that only samples with large fresh frozen material that meet a certain criteria could be included- something that needs a word or two. Tumor size has meaning and implications for metastases and outcomes.

Response: Please see the response to 2.2.

- 2.7 This reviewer would like a comment about coverage of WGS as this is a major economic issue that could be invoked with concerning consequences. The brief discussion about targeted sequencing, usually of high coverage, should at minimum address WGS coverage.

Response: We have expanded our discussion to comment on the coverage of WGS (please see the response to 2.2).

- 2.8 This reviewer agrees with the author's response to some of fellow reviewer #3's comments on access- this is a data set that can be reached but its process is arduous and the ability compute over the data set inconvenient but not impossible. The authors are accurate and this reviewer wishes that this discussion could lead to a more useful accession policy.....

Response: We thank the reviewer for recognising the issues relating to data accessibility associated with Genomics England. For a detailed reply, please see the response to 1.2.

Decision Letter, first revision:

13th Mar 2024

Dear Dr Houlston,

Thank you for submitting your revised manuscript "Novel cancer driver genes and opportunities for precision oncology revealed by analysis of 10,478 cancer genomes" (NG-A63519R1). It has now been seen by the original referees and their comments are below. The reviewers find that the paper has improved in revision, and therefore we'll be happy in principle to publish it in Nature Genetics, pending minor revisions to satisfy the referees' final requests and to comply with our editorial and formatting guidelines.

Sincerely,

Safia Danovi, PhD
Senior Editor, Nature Genetics

ORCID: 0009-0007-7822-5479

Reviewer #1 (Remarks to the Author):

Thank you for the additional context around data availability. This reviewer hopes that in the future there might be a reviewer link to access code and data to reproduce analyses in the manuscript, in a similar manner as GEO and dbGAP provide reviewer links. It is noted that this may be difficult across country borders and would likely require a signed waiver, but perhaps in the future, this could be possible.

"Across all cancers, we identified 330 cancer driver genes, 74 of which are novel to any cancer type."

Since there are no molecular validation studies included in this manuscript, this reviewer requests statements such as these to be reworded as "potential", "predicted", or "putative" cancer driver genes.

"Furthermore, 92% of the patients had self-reported European ancestry."

This statement should use either race or ethnicity as self-reported constructs, which are different concepts from genetic ancestry, and use the OMB categories ("White" for race or "Not Hispanic or Latino" for race). I point to a very important recent blog post following a publication interchanging these terms (<https://liorpachter.wordpress.com/2024/02/26/all-of-us-failed/>) which has been very helpful to this reviewer and our group.

Reviewer #2 (Remarks to the Author):

The revised manuscript is superb and now addresses the key issues as well as provides a clear and useful context for the extensive analyses. The authors are to be congratulated on this.

there are no major concerns here but just a few minor points to consider:

1. Novel in the title is challenging- might want to rethink this word
2. paragraph 2 page 3- "instead of organ site" should be further qualified: including stage and subtype.....
3. paragraph 3 page 3 might benefit from a comment on RNA and epigenetic analyses- as there are therapies now driven by both- esp the latter
4. last paragraph page 7...stomach cancers are notoriously heterogeneous and no two share similar karyotypes much less mutational profiles.....so underlying chaos might be cited...
5. in the discussion- page 16 might want to comment on trade-off of targeted (NHS and US based tests) vs WGS. also comment briefly on coverage?

As you can see the above are minor issues for consideration but all in all, now a very strong and important paper.

Stephen Chanock

Final Decision Letter:

1st May 2024

Dear Dr Houlston,

I am delighted to say that your manuscript "Analysis of 10,478 cancer genomes identifies candidate driver genes and opportunities for precision oncology" has been accepted for publication in an upcoming issue of Nature Genetics.

Your paper will be published online after we receive your corrections and will appear in print in the next available issue. You can find out your date of online publication by contacting the Nature Press Office (press@nature.com) after sending your e-proof corrections.

Acceptance is conditional on the data in the manuscript not being published elsewhere, or announced in the print or electronic media, until the embargo/publication date. These restrictions are not

intended to deter you from presenting your data at academic meetings and conferences, but any enquiries from the media about papers not yet scheduled for publication should be referred to us.

Please note that *Nature Genetics* is a Transformative Journal (TJ). Authors may publish their research with us through the traditional subscription access route or make their paper immediately open access through payment of an article-processing charge (APC). Authors will not be required to make a final decision about access to their article until it has been accepted. Find out more about Transformative Journals

Authors may need to take specific actions to achieve compliance with funder and institutional open access mandates. If your research is supported by a funder that requires immediate open access (e.g. according to Plan S principles) then you should select the gold OA route, and we will direct you to the compliant route where possible. For authors selecting the subscription publication route, the journal's standard licensing terms will need to be accepted, including [a href="https://www.nature.com/nature-portfolio/editorial-policies/self-archiving-and-license-to-publish"](https://www.nature.com/nature-portfolio/editorial-policies/self-archiving-and-license-to-publish). Those licensing terms will supersede any other terms that the author or any third party may assert apply to any version of the manuscript.

If you have not already done so, we invite you to upload the step-by-step protocols used in this manuscript to the Protocols Exchange, part of our on-line web resource, natureprotocols.com. If you complete the upload by the time you receive your manuscript proofs, we can insert links in your article that lead directly to the protocol details. Your protocol will be made freely available upon publication of your paper. By participating in natureprotocols.com, you are enabling researchers to more readily reproduce or adapt the methodology you use. [Natureprotocols.com](http://natureprotocols.com) is fully searchable, providing your protocols and paper with increased utility and visibility. Please submit your protocol to <https://protocolexchange.researchsquare.com/>. After entering your nature.com username and

password you will need to enter your manuscript number (NG-A63519R2). Further information can be found at <https://www.nature.com/nature-portfolio/editorial-policies/reporting-standards#protocols>

Sincerely,

Safia Danovi, PhD
Senior Editor, Nature Genetics
ORCID: 0009-0007-7822-5479